# Direct mapping of kidney function by DCE-MRI urography using a tetrazinanone organic radical contrast agent

Nicholas D. Calvert [1], Alexia Kirby[2], Mojmír Suchý[1], Peter Pallister[3], Aidan A. Torrens[1], Dylan Burger [4], Gerd Melkus [5,6], Nicola Schieda[6] & Adam J. Shuhendler [1,2,7] ✉

Chronic kidney disease (CKD) and acute kidney injury (AKI) are ongoing global health burdens. Glomerular filtration rate (GFR) is the gold standard measure of kidney function, with clinical estimates providing a global assessment of kidney health without spatial information of kidney- or region-specific dysfunction. The addition of dynamic contrast enhanced magnetic resonance imaging (DCE-MRI) to the anatomical imaging already performed would yield a 'one-stop-shop' for renal assessment in cases of suspected AKI and CKD. Towards urography by DCE-MRI, we evaluated a class of nitrogen-centered organic radicals known as verdazyls, which are extremely stable even in highly reducing environments. A glucose-modified verdazyl, glucoverdazyl, provided contrast limited to kidney and bladder, affording functional kidney evaluation in mouse models of unilateral ureteral obstruction (UUO) and folic acid-induced nephropathy (FAN). Imaging outcomes correlated with histology and hematology assessing kidney dysfunction, and glucoverdazyl clearance rates were found to be a reliable surrogate measure of GFR.

Chronic kidney disease (CKD) continues to be a major international healthcare burden, with the global mean prevalence > 13%[1,2]. CKD often develops slowly and without obvious symptoms in the early stages, but becomes progressively more debilitating in later stages with limited chances for reversal[2]. This disease is often attributed to long-term diabetes or hypertension, but is also a potential outcome following an acute kidney injury (AKI), the result of a sudden and dramatic decline in kidney function[3,4].

CKD outcomes are improved with early interventions, such as renin-angiotensin system blockade[5–7] or sodium-glucose co-transporter 2 (SGLT2) inhibitors[8] necessitating earlier detection[9]. Clinical diagnosis of CKD in North America is defined by proteinuria (albumin-to-creatinine (ACR) > 30 mg/g for >3 months) and/or functional decline (estimated glomerular filtration rate (eGFR) < 60 ml/min/1.73 m² for >3 months)[9,10]. These diagnostic values have been derived from large clinical studies in an ethnically-biased population, significantly reducing the diagnostic power of these biomarkers of disease[11]. Additionally, underlying causes of CKD can vary between individuals, where some of the most common, including diabetes mellitus, cardiovascular disease, and kidney transplant, can limit the accuracy of eGFR measurements at the patient level, as almost 30% of these patients can present with a 30% deviation from their true eGFR[12]. These issues may be partially mitigated by incorporation of cystatin C into the equation, however eGFR equations also assume steady-state creatinine and cystatin C levels and do not account for alterations in or alternate routes of creatinine production, exacerbating variability[13].

[1]Department of Chemistry and Biomolecular Sciences, University of Ottawa, 150 Louis Pasteur Pvt., Ottawa, Ontario K1N 6N5, Canada. [2]Department of Biology, University of Ottawa, 150 Louis Pasteur Pvt., Ottawa, Ontario K1N 6N5, Canada. [3]Department of Chemistry, Carleton University, 1125 Colonel By Dr., Ottawa, Ontario K1S 5B6, Canada. [4]Kidney Research Center, Ottawa Hospital Research Institute, University of Ottawa, 501 Smyth Rd, Ottawa, Ontario K1H 8L6, Canada. [5]Dept. Medical Imaging, The Ottawa Hospital, 501 Smyth Rd, Ottawa, Ontario K1H 8L6, Canada. [6]Dept. Radiology, University of Ottawa, 501 Smyth Rd, Ottawa, Ontario K1H 8L6, Canada. [7]University of Ottawa Heart Institute, 40 Ruskin St., Ottawa, Ontario K1Y 4W7, Canada. ✉e-mail: adam.shuhendler@uottawa.ca

It has become important that more accurate methods for GFR estimation be developed independent from demographic characteristics of the patient[14]. In addition, eGFR values fail to provide physicians with spatial or structural information underlying the renal dysfunction, omitting important diagnostic information for patients with AKI, CKD, and renal malignancies[15]. Kidney biopsy can provide histopathological data predictive of CKD outcomes, delivering spatial data about specific kidney lesions and not just overall kidney function[16]. However, biopsies are invasive procedures with their own inherent risk, precluding their repeated use to spatiotemporally characterize kidney disease. Clinically, GFR is the gold standard as an indicator for kidney function, however few clinical techniques are capable of providing spatial information about kidney function[17].

Medical imaging has been investigated as an alternative approach to measure kidney function. Imaging-based GFR assessment is of particular interest because it allows for a direct link between structural alterations (i.e. renal artery stenosis, ureteral obstruction) and changes in kidney filtration. In addition to this, GFR may be measured at the single kidney level. This is of particular interest when monitoring patients after partial nephrectomy or after living kidney donation where early increases in single kidney GFR are predictive of beneficial outcomes[18]. Importantly, imaging-based urographic approaches to GFR measurement mitigate errors associated with plasma sampling-based techniques, including sample timing, decay correction, dilution of standards and the handling of small volumes[19]. Camera-based imaging of GFR is, however, currently limited to $^{99m}$Tc-diethylenetriaminepentaacetic acid (DTPA) single photon emission computed tomography (SPECT)[14]. The specificity and sensitivity of camera-based GFR as diagnostic for renal failure was 100% and 47.5% for $^{99m}$Tc-DTPA, respectively[20]. $^{99m}$Tc-mercaptoazyltriglycerine (MAG3) is another SPECT-based method for imaging-based urography that is primarily cleared by tubular secretion, and is often used to evaluate renal plasma flow[14]. However, clearance of this radiotracer corrected for body surface area correlates well with creatinine-based GFR measurements[21]. These nuclear imaging techniques suffer from two sources of error leading to a wider confidence interval of the determined GFR relative to eGFR: background subtraction necessary for the correct measurement of percentage injected dose, and the estimation of renal depth from a population-derived nomogram based on patient height and weight in order to correct for signal attenuation[19,22]. Additionally, SPECT is also associated with limited spatial resolution and structural detail, however which is achievable by other imaging modalities such as magnetic resonance imaging (MRI)[23,24]. Dynamic contrast enhanced (DCE) MRI has been of particular interest as a tool for evaluating GFR in patients with renal artery stenosis, urinary obstruction, and living kidney donors. This technique gives spatial and structural reference to important clinical parameters that can be quantified through contrast enhancement changes over time, such as rate of contrast clearance and time to peak intensity[25,26]. In comparison to other medical specialties, the advances in imaging technique development in clinical nephrology have been slow.

The greatest challenge for kidney DCE-MRI is overcoming both the real and perceived risks associated with the use of traditional gadolinium (Gd)-based MRI contrast agents in patients with suspected or diagnosed renal dysfunction[27–30]. Contrast enhanced-MRI in patients with AKI and severe CKD following typical eGFR screens may be delayed or denied in patients with suspected kidney disease due to concerns of nephrogenic systemic fibrosis (NSF)[27,29]. NSF is a debilitating and sometimes fatal syndrome with no known treatment, and while its mechanism for pathogenic initiation is still poorly understood, it was shown to be linked to the accumulation of Gd in kidney tissue[27,29,31,32]. There is an increased risk of NSF in patients with severe renal dysfunction (AKI, dialysis patients, and stage 5 CKD) where lower rates of Gd clearance lead to higher residence times of the contrast agents[33]. This risk appears to be mitigated in group II Gd-based contrast agent (GBCAs), as defined by the American College of Radiology[34].

Regardless of the Gd-based contrast agent used, even with macrocyclic agents being more stable to linear chelators, there is a risk that low levels of gadolinium can be retained, for example at ppm-levels as phosphonates in the brain, and be slowly cleared over months to years[35,36]. Considering there are 30 million doses of Gd-based contrast agents administered annually worldwide, there is a need for a metal-free alternative for facilitating MRI-based urography[37]. There has been an effort to improve MRI contrast agent safety over the last decade by focusing on chemistries capable of altering water proton relaxation kinetics without the use of coordinated metal ions[38]. A significant proportion of these efforts has been focused on creating stable nitroxy radicals lacking α-protons in order to enhance their stability (i.e. 2,2,6,6-tetramethylpiperidine 1-oxyl, TEMPO). In addition to the cytotoxicity associated with free radical molecules, these organic radical contrast agents (ORCAs) are prone to reduction in vivo, quickly losing their radical character and contrast enhancing effects[39]. Efforts to stabilize these radicals with bulky sidechains and/or binding to metal nanoparticles significantly alter their biodistribution and clearance, complicating clinical translation[40,41].

Our group has taken an alternative approach to developing ORCAs, focusing on the radicals belonging to the 6-oxoverdazyl family. These molecules provide the chemical flexibility required for ORCA optimization, possessing a synthetic route that allows for functionalization of the *1* and *5* positions, as well as the *3* position of the tetrazinanone ring with stabilizing and targeting moieties (Fig. 1)[42,43]. Through a combination of other previously reported methods, we have optimized and scaled-up the synthesis of a verdazyl-based ORCA containing *3*-glucosyl and *1,5-N,N*-isopropyl groups (**4**, Fig. 1). While the *3* position glucosyl improves solubility and biocompatibility, the incorporation of the anomeric carbon into the tetrazinanone ring precludes cell uptake through glucose transporters, since the anomeric oxygen is required for substrate interaction with most of the GLUT family[44,45]. We also demonstrate enhanced stability and cytocompatibility of glucoverdazyl relative to a nitroxy radical (TEMPO) and spin trapping agent, 5,5-dimethylpyrroline-*N*-oxide (DMPO). The delocalization of the radical throughout the nitrogen-centered

**Fig. 1 | Synthesis of glucoverdazyl (4).** General synthetic scheme with yields used to synthesize glucoverdazyl. Red numbering on the final compound indicates numbering of the ring.

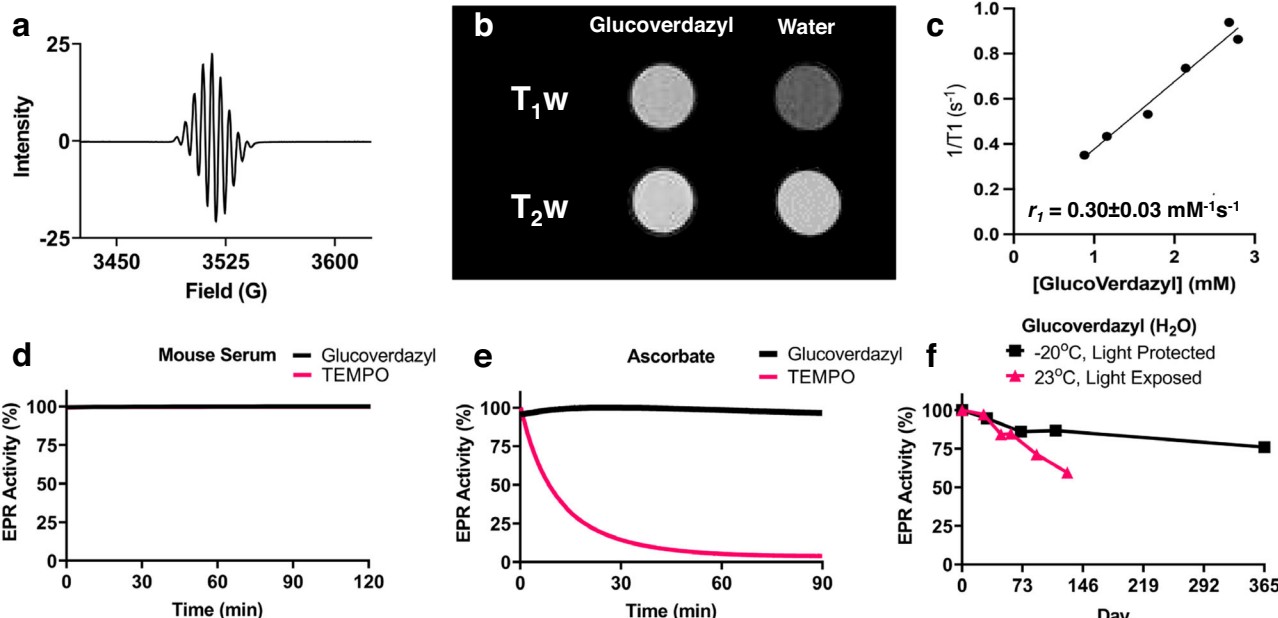

**Fig. 2 | Paramagnetic characteristics and stability of glucoverdazyl. a** EPR spectrum of a 5 mM glucoverdazyl solution in phosphate buffered saline (PBS) acquired at room temperature. **b** $T_1$ and $T_2$ weighted MRI images of 3 mM glucoverdazyl solution in PBS acquired at 3 T at 37 °C. **c** Longitudinal relaxivity of glucoverdazyl at pH 7.4 in PBS by 3 T MRI. The stability of glucoverdazyl (*black*) and TEMPO (*pink*) was determined by EPR during 2 h incubation in (**d**) mouse serum or (**e**) 4 mM sodium ascorbate buffer at pH 7.4. **f** The storage stability of 5 mM glucoverdazyl solutions left exposed to light at room temperature (*pink*) or left at −20 °C protected from light (*black*) as determined by EPR.

π-orbitals creates a more stable ORCA radical in relation to typical nitroxy radical agents[46]. We have evaluated the use of glucoverdazyl as a DCE-MRI agent through a short-term AKI and long-term AKI-to-CKD mouse model of kidney dysfunction (unilateral ureteral obstruction (UUO), and folic acid nephropathy (FAN), respectively), and have presented the DCE-MRI images as voxel-wise maps of glucoverdazyl perfusion to evaluate contrast clearance and retention. Finally, we compare the image-derived renal decay time constant of glucoverdazyl in healthy and disease mice to a validated transdermal method for GFR determination currently under clinical translation. With this comparison, we demonstrated that glucoverdazyl imaging can provide accurate and reproducible GFR maps from DCE-MRI, allowing the acquisition of kidney functional information. The present work lays the foundation for a unique class of ORCAs with the potential to provide for contrast enhanced-MRI in patients with renal dysfunction, and for the mapping of GFR onto the kidneys independent of the demographic characteristics of the patient.

## Results and discussion

### Optimized targeted synthesis of glucoverdazyl

A combination of previously reported 6-oxoverdazyl syntheses was used to identify an optimized route to glucoverdazyl yielding a high level of molecular purity and scalability that is required for an in vivo contrast agent (Fig. 1). The previously reported hydrazine side chains in the literature were usually limited to short carbon chains or aryl groups[44,47–50]. We functionalized the side chains with isopropyl groups, as they were a bulky enough side chain to help protect the delocalized radical while also improving serum retention after injection[51]. In earlier syntheses, we had generated *N*-Boc isopropyl hydrazine (**1**) from the *N*-Boc hydrazine precursor following a previously reported synthesis by Calabretta et al. in large quantities[52]. However, we found it more economical to purchase isopropyl Boc hydrazine. Compound **2** has been synthesized in a number of reported ways, all of which are di-substitutions of $COCl_2$, either as a phosgene solution or solid triphosgene[48,53]. While we have synthesized glucoverdazyl with both forms of phosgene with similar results, we have chosen to work with

the 15% phosgene in toluene solution. The route with the superior yield and purity was achieved through heptane recrystallization of the crude product after the phosgenation step, as first reported by Paré et al.[48]. Following Boc deprotection in ethanolic hydrochloric acid to form a crude intermediate used without further purification, we chose to generate the non-radical tetrazinanone ring (compound **3**) with D-glucose in the same manner as was first reported by Le et al.[42]. Finally, oxidation of compound **3** was also performed as reported by Le et al. using potassium ferricyanide, a much milder oxidant with an easier purification process compared to the more typically used benzoquinone seen in the majority of the available verdazyl literature[42,54–56]. Much of the past literature that has reported on these reactions have been incomplete with respect to characterization. Here we are reporting high purity $^1$H and $^{13}$C NMR spectra, as well as high-resolution mass spectra for each step and associated intermediates (Supplementary Figs. S2–S7). The purity of compound **4** was determined by EPR spectroscopy and analytical high-performance liquid chromatography, demonstrating that our approach to glucoverdazyl yielded **4** fully converted from non-radical **3** (Supplementary Fig. S8).

### Characterization of glucoverdazyl as an MRI-active contrast agent

The EPR spectrum for glucoverdazyl matched that reported previously, with the multiple EPR peaks being characteristic of radical delocalization within the tetrazinanone ring (Fig. 2a)[42,57]. The presence of the free radical indicated that glucoverdazyl would potentially result in MRI contrast enhancement[58]. MR imaging of solution phantoms of glucoverdazyl in PBS shows $T_1$ shortening but no change in $T_2$ relative to water (Fig. 2b). The relaxivity of glucoverdazyl was expectedly lower than that reported for gadolinium-based contrast agents, but of similar magnitude to other previously reported organic radical compounds, with a longitudinal relaxivity ($r_1$) of $0.30 \pm 0.03$ mM$^{-1}$s$^{-1}$ (Fig. 2c)[41,59,60]. In order to look at the possible mechanism of water relaxation for glucoverdazyl, the temperature-dependence of the corrected relaxation rate was examined (Supplementary Fig. S9). The small temperature-dependent increase in $R_2$ and lack of significant $T_2$-weighted image enhancement

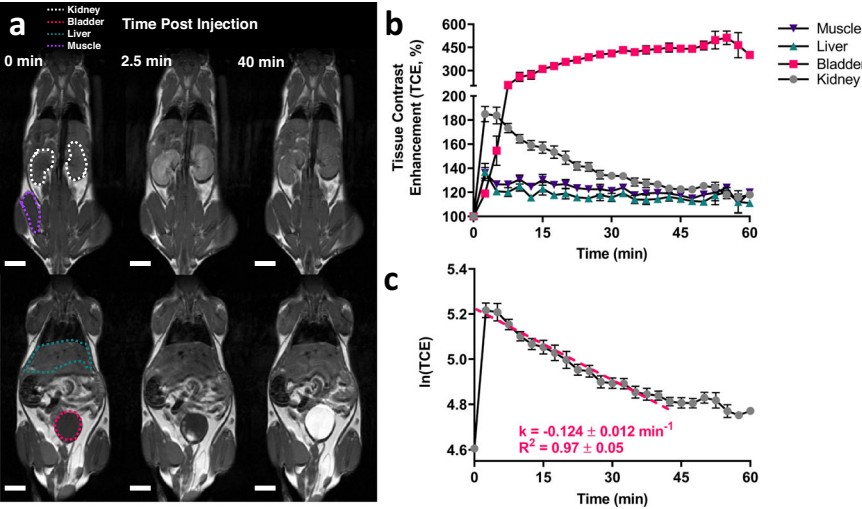

**Fig. 3 | Glucoverdazyl localization and clearance in vivo in healthy BALB/c mice. a** MRI scans of BALB/c mice were acquired both pre-injection and every 2.5 min post-injection following the administration of glucoverdazyl (3 mmol/kg). The white scalebar represents 5 mm. **b** Regions-of-interest (ROIs) were selected and the average intensity at each timepoint was acquired. Data are presented as mean ± SEM for $n = 9$ mice. **c** The semi-natural logarithmic transformation and line of best fit (pink) is drawn from $t = 2.5$ min to $t = 40$ min of the kidney clearance curve. Data are presented as mean ± SEM for $n = 9$ mice. The linear regression was determined individually for each mouse, and k was calculated as the slope of the fitted line. k and $R^2$ are shown as mean ± SD.

(Fig. 2b) suggests a negligible Curie spin relaxation mechanism[61]. The negligible Curie contribution is unsurprising as the spin quantum number (J or S) ($S = 1/2$) for ORCAs is lower than that for GBCAs, with electronic relaxation times ($T_{1e}$) being much larger for ORCAs (µs to high ns vs ps or less)[61,62]. Therefore, inner-sphere relaxation mechanisms would likely be dominated by Solomon-Bloembergen relaxation and chemical exchange[61,63]. Given the low relaxation rates of glucoverdazyl at high field (especially $R_2$), it is also possible there is an outer-sphere contribution to the relaxation rate as well[64].

While contrast enhancement was similar to TEMPO, the tetrazinanone radical present in glucoverdazyl was substantially more stable compared to the nitroxy ORCA counterpart (Fig. 2d, e). Neither glucoverdazyl nor TEMPO showed any change in radical persistence when incubated in mouse serum (Fig. 2d), however in the presence of ascorbate, a biological reductant, there was no loss of the glucoverdazyl radical but complete reduction of the TEMPO nitroxy radical within 90 min (Fig. 2e). Glucoverdazyl stability was also evaluated in the presence of hydrogen peroxide and glutathione (Supplementary Fig. S10), in the presence of Fenton chemistry (Supplementary Fig. S11), and superoxide generated from the xanthine-xanthine oxidase reaction (Supplementary Fig. S12). In all cases, glucoverdazyl maintained its radical character. In addition, glucoverdazyl was found to be stable at basic pH (i.e. pH 11) by both EPR and relaxation rate measurements, and demonstrating a small enhancement in relaxation rate at pH 3 ($\varepsilon^* = 1.33$ relative to pH 7) (Supplementary Fig. S13). The stability of the glucoverdazyl radical in solution was evaluated over time at room temperature in direct light or at −20°C in complete darkness. Periodic EPR sampling of these solutions showed that glucoverdazyl maintained over 50% of its radical character after 4 months being stored on the benchtop, while the frozen solution maintained over 80% of its radical activity after 1 year (Fig. 1f). This demonstrated the resiliency of the glucoverdazyl delocalized radical exposed to bioreductive conditions, as well as its shelf-life and storability.

The binding of glucoverdazyl to human serum albumin (HSA) at physiological levels was investigated in vitro in order to better define the putative mechanism of MRI contrast enhancement in vivo. The fractional binding of glucoverdazyl was evaluated (Supplementary Fig. S14A), demonstrating negligible glucoverdazyl binding to HSA at concentration below 1 mM and an increase in binding with [HSA] > 1 mM driven by mass action. These results were recapitulated by the

investigation of fluorophore displacement from HSA binding site I and site II (Supplementary Fig. S14B). Again, no binding at either site was noted below 1 mM HSA, with a slight increase in fluorophore displacement in both sites I and II suggestive of non-specific interactions driven by mass action. Finally, we evaluated the effect of the presence of HSA on $T_1$-shortening by glucoverdazyl (Supplementary Fig. S14C). The relaxation rate of a 0.1 mM and 1.0 mM glucoverdazyl solution was measured in the presence of HSA varying from 0% to 22.5% w/v in PBS. Similarly, to what was observed in the HSA binding experiments, there was very little change in glucoverdazyl relaxation rate at HSA concentrations at and below physiological levels (i.e. [HSA] <1 mM), before mass action-induced non-specific binding occurred. These data suggest that glucoverdazyl will not interact with serum albumin once injected *intravenously*.

The cytocompatibility of glucoverdazyl was evaluated in H460 human lung carcinoma epithelial cells and demonstrated no cytotoxicity compared to untreated cells at concentrations up to 10 mM (Supplementary Fig. S15). Glucoverdazyl was evaluated for contrast media suitability in vivo following *intravenous* injections to 9 BALB/c mice. An administered dose of 3 mmol/kg was chosen based on the difference in $r_1$ between glucoverdazyl and Gadovist® (~10-fold), and considering the standard clinically recommended Gadovist® dose of 0.1 mmol/kg. This dose of glucoverdazyl was still well below the maximum concentration we evaluated for cytocompatibility. Following injection, $T_1$-weighted images were acquired every 2.5 min following a pre-injection scan, which was used to establish baseline voxel intensity. Limited contrast enhancement was observed in the muscle and liver, with uptake and clearance isolated to the urinary system (Fig. 3a). Overall, contrast enhancement relative to pre-contrast scans reached 127 ± 9% in the muscle, a 121 ± 10% in the liver, and 184 ± 21% in the kidneys 5 min post injection (Fig. 3b). The average clearance time after injection, determined by the return of the kidney ROI to baseline intensity, was ~40 min, coinciding with the plateau of the bladder ROI signal increase. At this timepoint, we observed contrast enhancement of 120 ± 8% in the muscle, 115 ± 7% in the liver, 127 ± 8% in the kidneys, and 438 ± 48% in the bladder. The clearance kinetics of glucoverdazyl through the kidneys matched that of one-phase decay, affording the determination of the renal decay time constant (RDTC, k in $min^{-1}$) from the slope of the semi-natural log plot of the data (Fig. 3c). Linear regression was applied from the contrast intensity maximum at

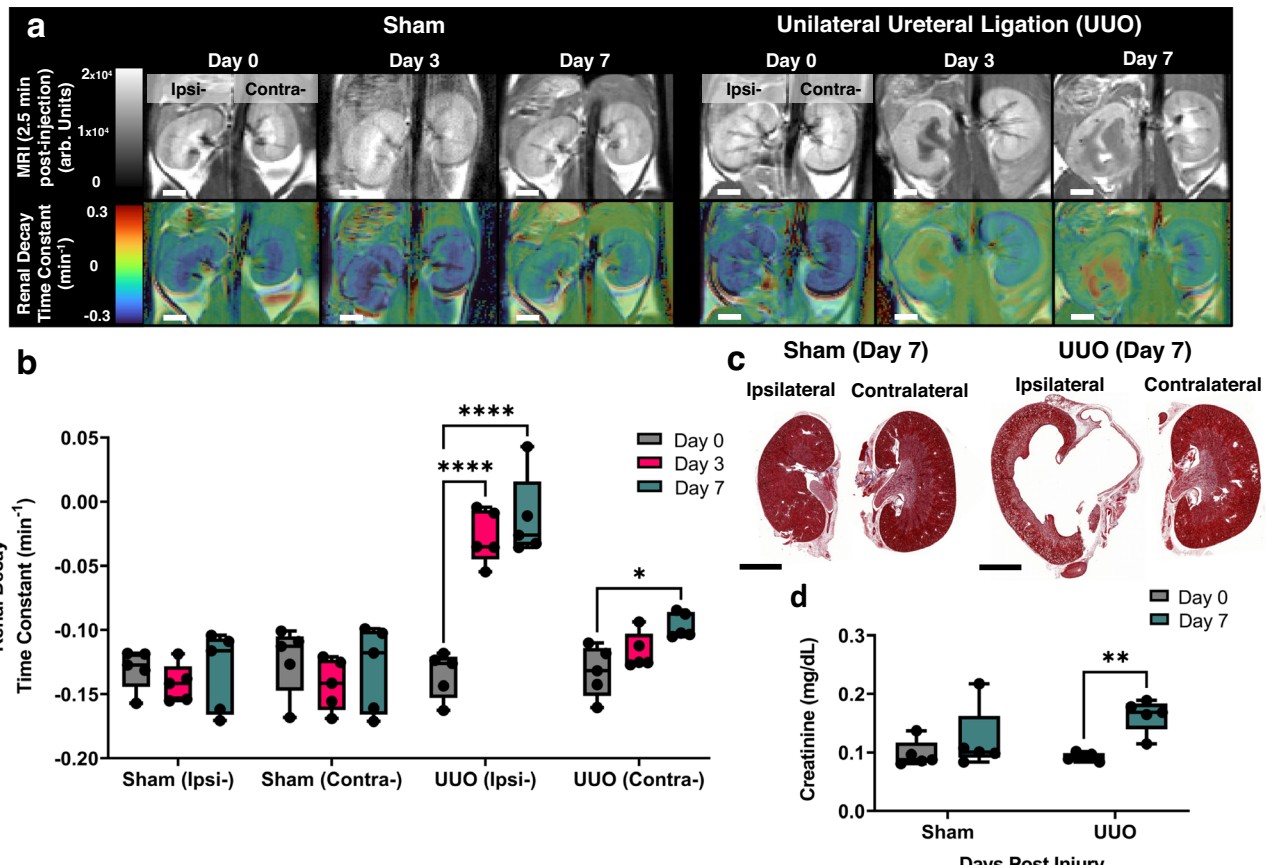

**Fig. 4 | Glucoverdazyl-enhanced DCE-MRI in a mouse mode of unilateral ureteral obstruction. a** $T_1$-weighted images of kidneys at $t = 2.5$ min post-injection (*top*) and RDTC maps (*bottom*) for Sham and UUO groups. The white scalebar represents 2 mm. **b** RDTC values for each kidney at each post-injury time-point. Data are presented as box-and-whisker plots of the single average RDTC value from each kidney (ipsi- or contralateral) individually, from each mouse ($n = 5$). **c** Representative histology selected from one of the five paired mouse kidneys from sham (*left*) and UUO (*right*) treatment groups stained by PAS. The black scalebar

represents 6 mm. **d** Serum creatinine levels of sham (*grey*) and UUO (*turquoise*) mice on the two sampled days. Data are presented as boxplots individual values of SCr. Statistical analysis was done by repeated measures two-way ANOVA followed by a Tukey post-hoc test. In all graphs, *$p = 0.019$, **$p = 0.003$, and ****$p < 0.0001$. In all boxplots, whiskers are drawn from the minimum to maximum values, box bounds represent the interquartile range, and the line within the box represents the median.

$t = 2.5$ min to the timepoint of the most consistent return to baseline at $t = 40$ min. The k determined from the average data from 9 healthy BALB/c mice was $-0.124 \pm 0.012$ min$^{-1}$, with an $R^2$ of $0.97 \pm 0.05$, demonstrating excellent reproducibility of this baseline measure of healthy kidney function.

With the observation that glucoverdazyl was being primarily taken up by the kidneys, cytocompatibility studies were repeated with high glucoverdazyl concentrations in human renal proximal tubule cells (hRPT, Supplementary Fig. S16). While glucoverdazyl resulted in no significant increase in cell death compared to untreated cells after a 24 h incubation, >90% of the hRPT were dead after only a 4 h incubation with TEMPO. This stark difference in cytocompatibility highlights an additional key performance difference between TEMPO and verdazyl-derived ORCAs. The cell uptake of glucoverdazyl by hRPT cells and glucose-starved HepG2 cells was evaluated by EPR spectroscopy, and indicated that glucoverdazyl was not taken up in any detectable manner (Supplementary Fig. S17). Overall, low extraurinary glucoverdazyl biodistribution, the relatively fast renal clearance time of glucoverdazyl, and its demonstrated cytocompatibility suggest that glucoverdazyl may be well suited to MRI-based evaluation of kidney function.

### Glucoverdazyl-DCE-MRI of acute kidney injury through unilateral ureteral obstruction

A unilateral ureteral obstruction (UUO) mouse model was used to determine the effectiveness of glucoverdazyl as a DCE-MRI agent for

an AKI caused by obstructive nephropathy. The surgical obstruction of the left ureter prevents fluid clearance, leading to hydronephrosis and dramatically reduced kidney function of the ipsilateral kidney. Both sham (left kidney was touched by a surgical instrument) and surgical UUO (left kidney was ligated) mouse groups were evaluated. The change in voxel-wise intensity over time within the kidneys was determined for the entire left and right kidney (Fig. 4a, *top panel*), as well as their cortex and medullary/renal pelvis (MRP) regions (Supplementary Fig. S18). Voxel-wise mapping of the RDTC (Fig. 4a, *bottom panel*) was used to evaluate changes in kidney function. Hydronephropathy was evident by the ablation of the medullary region on day 3, which continued to worsen at day 7, a hallmark of the UUO model[65]. The kidney contralateral to the ligated ureter in the UUO mice did not show any obvious morphological changes.

The RDTC was determined in C57/Bl6 mice prior to surgery in order to define the optimal time interval for glucoverdazyl clearance prior to the analysis of data from diseased mice (Supplementary Fig. S18). In the sham-treated mice, no significant changes in RDTC were noted (Fig. 4b) on days 3 or 7 relative to day 0 across both kidneys. In UUO-treated mice, however, a significant change in RDTC of the ipsilateral kidney was observed, increasing from $k = -0.135 \pm 0.018$ min$^{-1}$ on day 0 to $k = -0.028 \pm 0.014$ min$^{-1}$ on day 3, and $k = -0.013 \pm 0.032$ min$^{-1}$ day 7. The RDTC of the contralateral kidney in UUO mice was unchanged between days 0 and 3 ($k = -0.133 \pm 0.020$ min$^{-1}$ versus $k = -0.117 \pm 0.014$ min$^{-1}$, respectively), but showed a significant increase

in RDTC by day 7 ($k = −0.097 ± 0.010$ min$^{-1}$). Altered physiology of the contralateral kidney following UUO is expected in rodent models, with the induction of macrophage-to-myofibroblast transition[66], fibrosis[66,67], and altered cortical mitochondrial function[68] previously reported. Our data demonstrate that glucoverdazyl-mediated DCE-MRI was able to detect the contralateral kidney functional impairment early after ipsilateral ureteral obstruction.

In comparison to the RDTC, mapping area under the curve (AUC) was unreliable for evaluating kidney function using glucoverdazyl (Supplementary Figs. S18, S19). AUC relies heavily on the raw intensity values of the kidney voxel during each scan, and is the average outcome over uptake, retention, and excretion processes. As such, alterations in AUC may indicate abnormalities, but with the underlying deficiency remaining indiscernible across contributing processes. In both the sham and UUO-treated mice, no significant difference in AUC was found between day 0 and any other of the post-injury time points in either the contralateral or ipsilateral kidneys, except for a significant decrease in AUC between days 3 and 7 observed in sham mice. Overall, the alterations in kidney function revealed by RDTC mapping were not recapitulated in the AUC-derived analyses. Histological and serum creatinine (SCr) analysis was performed as gold standard measures to corroborate pathology we observed through image-based functional evaluation. Both the morphology and fibrosis staining of the sham kidneys and the ipsilateral kidney in UUO-treated mice were unremarkable, while the kidney ipsilateral to ureteral obstruction clearly shows hydronephropathy (Fig. 4c). No significant change in SCr was detected in the sham mice between days 0 and 7 post-surgery, while a significant increase is observed in the UUO mice (Fig. 4d). The observed changes in SCr align well with those previously reported in UUO models, and recapitulate the observed alterations in RDTC[69–71]. The fact that AUC measurements do not parallel the measured alteration in SCr supports the use of RDTC as a measure of kidney function by glucoverdazyl-mediated DCE-MRI.

While the damage caused to the kidney ipsilateral to the ureteral obstruction is evident even on anatomical MRI, the UUO model demonstrated that standard glucoverdazyl-mediated DCE-MRI techniques with simple kinetic mapping can be used to show regional and structural defects across both kidneys, highlighting the functional changes arising in the contralateral kidney even before positive fibrosis staining. A regional analysis of the data discriminating glucoverdazyl clearance from the cortex *versus* medulla & renal pelvis was performed (Supplementary Fig. S18). Here, we are able to clearly discern an impairment in glucoverdazyl clearance caused by a decrease in kidney function, while also mapping where the pathological process is taking place within the kidney, which is valuable towards the evaluation of AKI[44–46].

## Glucoverdazyl-DCE-MRI of acute-to-chronic kidney injury through folic acid-induced nephropathy (FAN)

We next sought to evaluate renal function in a more complex, fibrosis-driven model of kidney disease mediated through folic acid-induced nephropathy (Fig. 5). FAN is the result of tubular folic acid crystal formation following systemic administration of folic acid[72]. This crystallization causes an initial phase of severe AKI, which is followed by fibrotic renal scarring. The fibrotic scars lead to long term, progressive decline in kidney function, resulting in CKD around 3-weeks post-folate injection. The FAN model was implemented in BALB/c mice rather than the more commonly used C57Bl/6 strain, as BALB/c mice were more resistant to the AKI phase, which had a very high mortality rate in C57Bl/6 mice.

Through anatomical imaging, we noted a reduction in overall kidney size from day 0 to day 30 in both kidneys, which has previously been reported for the FAN model[73] (Fig. 5a). Following glucoverdazyl-mediated DCE-MRI, the RDTC was determined for both kidneys prior to the administration of folate, and 15- and 30-days

following administration (Fig. 5a, b). A significant increase in RDTC was observed at day 15 ($k = −0.154 ± 0.025$ min$^{-1}$ on day 0 and $k = −0.082 ± 0.008$ min$^{-1}$ on day 15), followed by a return to baseline RDTC at day 30 ($k = −0.139 ± 0.016$ min$^{-1}$). The AKI phase of FAN resulted in a significant increase in medullary and cortical RDTC, indicative of both, poor drainage into the ureter and poor glomerular filtration. The recovery of RDTC by day 30 is anticipated in the early stages of the CKD phase of the disease since AKI often presents with much more severe kidney dysfunction than early stages of CKD, which agreed with the literature using the same post-injury time points[73]. The RDTC maps at day 30 indicate the presence of localized, striated regions within the cortex presenting relatively slower clearance rates (Fig. 5a *right*, black arrows), recapitulating the fibrotic striations observed in kidney histology at day 30 (Fig. 5c, light blue regions are indicators of fibrotic tissue highlighted by black arrows). Therefore, the combination of spatial and temporal information into single kidney maps, as is done in the case of RDTC images (Fig. 5a), afford an increased diagnostic power not afforded by existing nephrological techniques limited either to spatial or kinetic information alone. Importantly, the AUC maps do not provide the same correlative structure-function detail that was obtained from RDTC (i.e. no striations as a result of fibrosis), indicating that AUC is an underpowered method through which kidney function can be evaluated (Supplementary Fig. S21). Data segmented into cortex or medulla & renal pelvis regions were also evaluated (Supplementary Fig. S20).

Kidneys were harvested at days 0, 15, and 30 post-folate injection and evaluated by histology to confirm both AKI and CKD (Fig. 5c), and blood sampling was performed over the same time intervals for SCr determination (Fig. 5d). Histological evaluation revealed substantially increased fibrotic regions in the kidney at day 15, which decreased in severity at day 30 (Fig. 5c). SCr showed a slight elevation on day 15 relative to day 0, which was expected as AKI usually shows only small elevations in SCr[73,74]. However, by day 30 during early stages of CKD, SCr was significantly increased compared to both day 0 and day 15, which is a strong indicator of a severe decrease in kidney function associated with FAN and early stages of CKD[75–77]. Overall, the importance of combining spatial and temporal dimensions in urography was highlighted by the assessment of the FAN model. During severe AKI, minor elevations in SCr were seen, but a much larger increase in RDTC was obtained with differential regional effects across the kidney (Fig. 5a). Of unique value, RDTC maps indicated discrete regions of greater kidney dysfunction (i.e. cortical striations), which may correspond to areas of fibrosis observed in the histological evaluations (Fig. 5c).

## Comparison of glucoverdazyl DCE-MRI to a validated measure of GFR

To further develop glucoverdazyl-mediated DCE-MRI as an imaging-based approach to renal functional evaluation, we compared image-derived RDTC values to an established and validated measure of GFR: transdermal fluorescence monitoring[64–67]. The transdermal GFR technique relies on the *intravenous* injection of a fluorescent molecule (i.e. FITC-sinistrin) cleared solely by filtration, and the transdermal monitoring of blood-pool fluorescence over time. The transdermal technique applies a one-phase decay model to determine the RDTC of the fluorescence intensity *versus* time curve, which is then corrected to GFR by a previously determined correction factor[78–81]. We used the transdermal GFR measurements as a benchmark against which the glucoverdazyl-specific correction factor could be derived.

In a second cohort of FAN-induced mice, both transdermal (Fig. 6a) and glucoverdazyl-mediated DCE-MRI (Fig. 6b) measurements were performed on day 0 (black), day 15 (pink) and day 30 (cyan). Both techniques resulted in the hallmark one-phase decay curve, yielding RDTC values after semi-natural log transformations. On day 0, the mean RDTC determined by glucoverdazyl-mediated DCE-MRI ($k = −0.135 ± 0.022$ min$^{-1}$) was significantly different to the RDTC

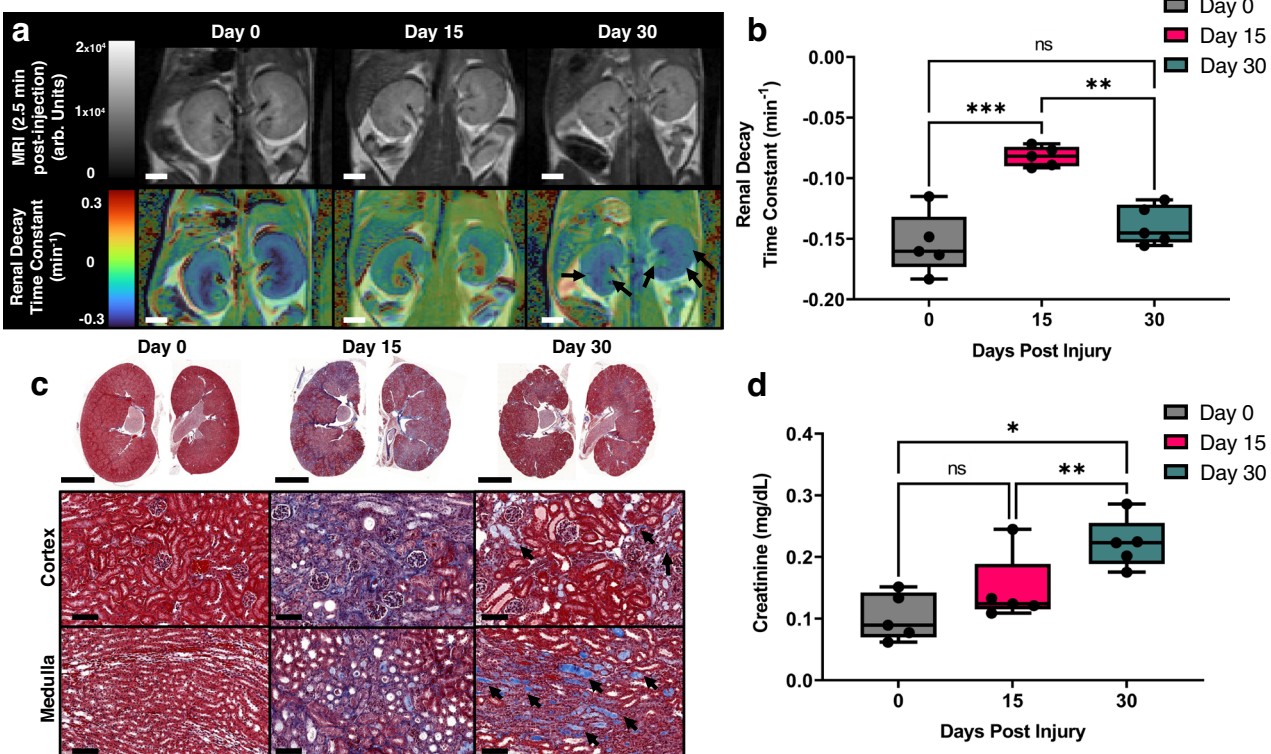

**Fig. 5 | Glucoverdazyl-enhanced DCE-MRI of folic acid-induced nephropathy.**
**a** $T_1$-weighted images of kidneys at $t = 2.5$ min post-injection (*top*) and RDTC maps (*bottom*). The white scalebar represents 2 mm. **b** RDTC values for kidneys at each post-injury time point. Data are presented as box-and-whisker plots of the single average RDTC value from both kidneys of each mouse ($n = 5$). **p = 0.004 and ***p = 0.001. **c** Representative histology selected from one of the five paired mouse kidneys (*top*) with enlargements of the cortical or medullary regions (*bottom*) stained by PAS. The top black scalebar represents 6 mm and the bottom black scalebar represents 100 μm. Black arrows indicate positive histological staining for fibrotic areas, represented by a light blue color. **d** Serum creatinine levels of FAN mice at each post-injury time point ($n = 5$). *p = 0.025 and **p = 0.004. Statistical analysis was done by repeated measures two-way ANOVA followed by a Tukey post-hoc test. In all boxplots, whiskers are drawn from the minimum to maximum values, box bounds represent the interquartile range, and the line within the box represents the median.

determined by transdermal fluorescence ($k = -0.075 \pm 0.011$ min$^{-1}$) (Supplementary Fig. S16). The difference in these RDTCs is unsurprising given the different locations for data sampling. While the transdermal measurements evaluate the signal from blood pool only within the first few millimeters under the surface of the skin, the DCE-MRI technique directly evaluates kidney tissue. By applying the transdermal technique, baseline GFR = 1584 ± 238 μl/min/100 g b.w. was determined for BALB/c mice, which is in line with literature values reported[82]. By pooling the baseline DCE-MRI-derived RDTC data obtained for day 0 BALB/c mice, and by using the average GFR measurement from the transdermal technique, we were able to derive a glucoverdazyl-specific correction factor for the conversion of RDTC to GFR. With this factor, we calculated GFR from our RDTC values for each post-injury time point and compared them to the GFR values determined by the transdermal technique (Fig. 6c), demonstrating that there was no significant difference in GFR determined by the two methods.

Deviations in measured GFR between the DCE-MRI and transdermal methods applied were observed (Fig. 6c), and these differences may derive from the very different locations for data sampling: The transdermal method relies on blood pool signal near the skin surface, where DCE-MRI evaluates contrast change within the kidneys directly. During severe AKI on day 15, RDTC is dictated entirely by the reduction in filtration as a result of AKI (Fig. 6d). On day 30, however, the major AKI has healed and early CKD as a result of fibrotic scarring has begun. The fibrotic scarring is more prevalent in the medulla (Fig. 5c), which means excretion dictates the value of RDTC. Because our MRI method spatially maps contrast agent clearance on the kidney itself, the differentiation of filtration-limited clearance *versus* excretion-limited

clearance is possible. Since the transdermal method measures agent in the blood pool (i.e. non-spatially encoded data), this distinction in reduced clearance mechanisms cannot be made. Overall, the data support the application of glucoverdazyl as a DCE-MRI contrast agent suitable for mapping GFR over the volume of the kidney.

We have optimized and scaled a preparation of a tetrazinanone-derived ORCA, glucoverdazyl, and demonstrated its superior redox stability and cytocompatibility relative to previously used nitroxy-radical contrast agents. Glucoverdazyl appears especially useful for renal DCE-MRI due to its biodistribution limited to the kidneys, ureter, and bladder. Glucoverdazyl has been applied to imaging the UUO model of severe AKI, and FAN model of AKI-to-CKD progression, showing regional functional changes within the kidneys in the form of RDTC. The evaluation of dynamic image sets using RDTC fully corroborated kidney dysfunction in both models. Through benchmarking to a validated transdermal fluorescent recording method of measuring GFR, we demonstrated that glucoverdazyl affords the reliable determination of GFR by DCE-MRI. Importantly, this approach to GFR measurement not only adds a spatial component to urography, but also determines GFR free from the reliance upon patient demographic characteristics, which has been shown to be error-prone. In order to overcome limitations of the current study, future improvements to RDTC measurement will include a higher frequency of imaging and volumetric image acquisition through pulse sequence development, which would increase the confidence and fidelity of our proposed approach to MRI urography. Overall, glucoverdazyl may provide safer MRI-based diagnoses in patients with known or suspected AKI and/or CKD. Given the molecular properties of this organic radical, especially

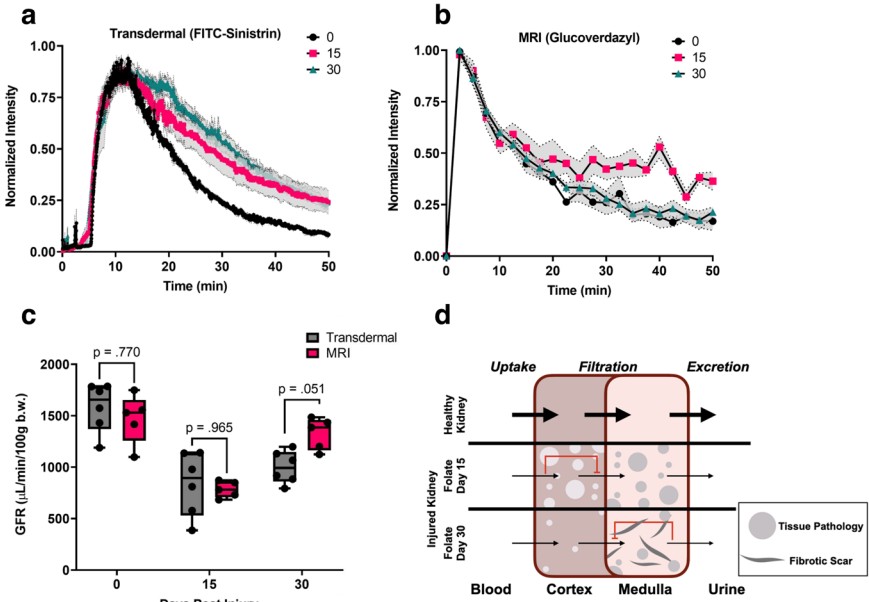

**Fig. 6 | Determination of glomerular filtration rate (GFR) by transdermal fluorescence and dynamic contrast enhanced magnetic resonance imaging (DCE-MRI) in folic acid nephropathic (FAN) mice. a** Normalized fluorescence intensity of the transdermal fluorescence clearance of FITC-sinistrin in FAN mice on day 0, 15, and 30 (black, pink, and turquoise, respectively). Data are presented as the mean of each time point for each replicate ± SEM for $n = 6$ mice. **b** Normalized DCE-MRI intensity of glucoverdazyl-enhanced scans over time for FAN mice on day 0, 15, and 30 (black, pink, and turquoise, respectively). Data are presented as the mean of each time point for each replicate ± SEM for $n = 5$ mice. **c** GFR values for

FAN mice determined by either transdermal fluorescence (grey) or DCE-MRI (pink). Data are presented as box-and-whisker plots of the GFR for each mouse. Statistical analysis was done by mixed measures two-way ANOVA followed by a Tukey post-hoc test. The *p-value* for each test is labelled on the graph. **d** Illustration of glucoverdazyl clearance through the kidney by DCE-MRI to elaborate on differences between transdermal and DCE-MRI measurements. In all boxplots, whiskers are drawn from the minimum to maximum values, box bounds represent the inter-quartile range, and the line within the box represents the median.

its inability to enter the cell, cytocompatibility and preliminary bio-compatibility, localization to the kidneys and quick clearance, tetrazinanone-based ORCAs represent an innovative and promising class of metal-free MRI contrast agent.

## Methods
All animal studies were conducted under Animal Use Protocol HIe-3640-R1 approved by the IACUC at the University of Ottawa.

### General reagents
All chemical reagents were purchased from Sigma-Aldrich and used as is unless otherwise reported, with the exception of *N'*-(propan-2-yl)(tert-butoxy)carbohydrazide, which was purchased from AABlocks. All solvents were HPLC grade, except for water (18.2 MΩ cm Millipore water).

### Experimental procedures
All NMR spectra were acquired on a Bruker AVANCE II 400 MHz or a Bruker Avance III HD 600 MHz NMR spectrometers, operating at 400 MHz or 600 MHz for $^1$H spectra, and 100 or 150 MHz for $^{13}$C spectra, respectively. Chemical shifts are given in ppm. In all spectra, CDCl$_3$ was referenced to 7.26 ppm ($^1$H NMR) and 77.0 ppm ($^{13}$C NMR); MeOH-D$_4$ to 3.31 ppm ($^1$H NMR) and 49.0 ppm ($^{13}$C NMR). All high-resolution mass spectrometry (HRMS) were performed on a Micro-mass Q-TOF I with electrospray ionization. All EPR spectra were acquired on a Bruker EMX plus EPR at room temperature. All MRI acquisitions were performed on a 3 T pre-clinical MRI scanner (MR Solutions, Ltd.). For all MRI image data analysis, only relevant slices of the tissue of interest were included in analysis (i.e., scans for kidney region only used slices with the kidney visible). All data processing, mapping, and quantity generation was done using a custom program written in MATLAB 2020 A®. A copy of this program is available from the authors upon request. GraphPad Prism 9.5 was used to generate all graphs, graphical figures, and statistical results.

### Synthesis of N-({N'-[(tert-butoxy)carbonyl]-N-(propan-2-yl)hydrazinecarbonyl}(propan-2-yl)amino)(tert-butoxy)formamide (2)
Dry Et$_3$N (4 ml, 28.8 mmol) was added to a solution of toluene (50 mL, pre-died with 4 Å molecular sieves) followed by the addition of 4 g of *N'*-(propan-2-yl)(tert-butoxy)carbohydrazide (4 g, 23 mmol). The solution was cooled to 0ºC while being stirred in the atmosphere of N$_2$. A 15% phosgene solution in toluene (9 mL, 12.7 mmol) was added dropwise for ~1 min (phosgene is highly toxic, caution is needed when performing the addition), and the reaction mixture was stirred for 1 hr at 0ºC, then warmed to room temperature (rt) and stirred for an additional 18 h. The reaction was quenched by adding MeOH (50 mL), was stirred for 30 min at rt and was evaporated. The mixture was diluted with 10% solution of NH$_4$OH (75 mL) and extracted with EtOAc (3 × 15 mL). The combined organic was washed with brine (40 mL), dried with Na$_2$SO$_4$, filtered, and evaporated, resulting in a white powder. The powder was dissolved in 80 mL of hot, dry heptane and set aside for 18 hrs at 4ºC to induce crystallization of the product. The crystals were filtered off and were washed with hexanes. The product was dried on the high vacuum, (colourless crystals, compound **2**, 2.55 g, 59%).

**$^1$H NMR** (400 MHz, CDCl$_3$) δ 6.40 (s, D$_2$O exch., 2H), 4.15 (s, 2H), 1.43 (s, 18H), 1.12 (s, 12H).

**$^{13}$C NMR** (150 MHz, CDCl$_3$) δ = 155.9, 81.1, 52.6, 50.3, 28.2, 19.2 (broad signal).

**HRMS (ESI):** Calculated for C$_{17}$H$_{34}$N$_4$O$_5$Na [M+Na]+: 397.2411, found 397.2427.

### Synthesis of 1,3-diamino-1,3-bis(propan-2-yl)urea dihydrochloride
Compound **2** (2.55 g) was resuspended in EtOH (25 mL) in a 100 mL round bottom flask and heated to 80ºC (air condenser). Concentrated HCl (10 mL) was added dropwise and the solution was left stirring for

30 min at 80 °C. The solution was cooled to rt and the solvent was evaporated. The crude product was consecutively co-evaporated once with methanol, toluene, and petroleum ether (50 mL for each). The crude product of sufficient purity for the subsequent step was dried on the high vacuum giving 1,3-diamino-1,3-bis(propan-2-yl)urea dihydrochloride as colourless solid in quantitative yield.

**$^{1}$H NMR** (400 MHz, MeOH-D$_4$) δ 4.23 (heptet, $J$ = 6.8 Hz, 2H), 1.35 (d, $J$ = 6.8 Hz, 12H).

**$^{13}$C NMR** (150 MHz, MeOH-D$_4$) δ = 162.3, 56.2, 18.9.

**HRMS (ESI)**: Calculated for $C_7H_{18}N_4ONa$ [M+Na]+: 197.1355, found 197.1378.

## Synthesis of 6-[(1 S,2 R,3 R,4 R)-1,2,3,4,5-pentahydroxypentyl]-2,4-bis(propan-2-yl)-1,2,4,5-tetrazinan-3-one (3)

1,3-Diamino-1,3-bis(propan-2-yl)urea dihydrochloride, (1.55 g, 6.3 mmol) was resuspended in H$_2$O (10 mL) with stirring at rt. A 5 mL solution of D-glucose (1.2 g, 6.7 mmol) and NaOAc (1.1 g, 13.4 mmol,) in water was added dropwise over 1 min, followed by stirring at rt for 18 hrs. The reaction mixture was extracted with $n$-butanol (6 × 10 mL). The combined organic extract was dried with Na$_2$SO$_4$, was filtered, and was evaporated. The resulting oil was consecutively co-evaporated with methanol, toluene, and petroleum ether (50 mL of each). The resulting product was dried on the high vacuum overnight and giving pale yellow crystals (compound **3**,1.48 g, 70%).

**$^{1}$H NMR** (400 MHz, MeOH-D$_4$) δ 4.53 (m, 2H), 4.06 – 3.95 (m, 2H), 3.84 – 3.70 (m, 2H), 3.68 – 3.57 (m, 2H), 3.53 (d, $J$ = 2.7 Hz, 1H), 1.13 (dd, $J$ = 6.8, 3.4 Hz, 6H), 1.08 (d, $J$ = 6.5 Hz, 6H).

**$^{13}$C NMR** (150 MHz, MeOH-D$_4$) δ = 155.7, 73.0, 72.4, 72.2, 72.0, 70.0, 65.0, 19.8, 19.7, 19.3, 18.9.

**HRMS (ESI)**: Calculated for $C_{13}H_{28}N_4O_6Na$ [M+Na]+: calculated 359.1910, found 359.1907.

## Synthesis of 3-oxo-6-[(1 S,2 R,3 R,4 R)-1,2,3,4,5-pentahydroxypentyl]-2,4-bis(propan-2-yl)-1,2,3,4-tetrahydro-1,2,4,5-tetrazin-1-yl (glucoverdazyl, 4)

Compound **3** (1.48 g, 4.41 mmol) was resuspended in H$_2$O (5 mL with stirring (rt)). In a separate vessel, potassium ferricyanide (4.44 g, 13.5 mmol) was mixed with 80 drops (~4.5 mL) of NaHCO$_3$ solution (2 M), followed by the addition of water (5 mL); the mixture was solubilized using an ultrasound bath. Resulting solution was added dropwise over 1 min to the original reaction mixture with stirring followed by stirring (rt) for about 30 min or until effervescence stopped. The mixture was extracted with $n$-butanol (6 × 10 mL). The combined organic was dried with Na$_2$SO$_4$, was filtered, and was evaporated. The resulting oil was consecutively co-evaporated with methanol (50 mL), three times with toluene (50 mL each time), was cooled to to 0 °C, followed by co-evaporation with petroleum ether (50 mL). The resulting product was dried on the high vacuum overnight to give a bright yellow fine powder (glucoverdazyl **4**,1.09 g, 74%). Given the compound was radical in nature it could not be characterized by NMR; changes in HPLC elution time, as well as HRMS and EPR were used to confirm structure and purity. HPLC traces can be seen in Supplementary Fig. S7.

**HRMS (ESI)**: Calculated for $C_{13}H_{25}N_4O_6Na$ [M+Na]+: calculated 356.1676, found 356.1672.

## Phantom MRI of glucoverdazyl

Glucoverdazyl was prepared in 1 × PBS in standard NMR tubes. These were inserted into a 50 mL Falcon tube containing ultrasound gel, comprising the MRI phantom. The MRI phantom was placed into a 38-mm-diameter send and receive volume coil and inserted into the MRI scanner. A multislice Rapid Imaging with Refocused Echoes (RARE) pulse sequence was implemented for evaluation of phantoms using the following parameters for $T_1$-weighted imaging: slice thickness of 5 mm, FOV of 40 × 40 mm, averages = 3, matrix size = 96 × 96, TE$_{eff}$ = 11 ms, echo spacing = 7 ms, TR = 720 ms, and acquisition time of

2 min 16 s. For $T_2$-weighted imaging, all parameters were the same as for $T_1$-weighted images, except TE$_{eff}$ = 68 ms and TR = 4800 ms, and acquisition time was 7 min 28 s.

For relaxivity measurements, the same imaging phantom was used with contrast agent concentrations of 1 to 3 mM. Contrast agent concentration following imaging was verified by electron paramagnetic spectroscopy. To measure the longitudinal relaxation rate (R$_1$), an inversion recovery RARE sequence was implemented with the following parameters: Slice thickness of 5 mm, FOV of 50 × 50 mm, average = 1, matrix size = 96 × 96, TE$_{eff}$ = 17 ms, TR = 5000 ms, TI = 50, 75, 100, 150, 200, 250, 300, 400, 600, 800, 1200, 2400, and 4800 ms, and acquisition time of 2 min 30 s $per$ TI. Longitudinal relaxation rates were extracted using the mapping MATLAB 2 routine written by J. Barral, M. Etezadi-Amoli, E. Gudmundson, and N. Stikov (2009), and modified by J. Rioux (2022). The longitudinal relaxivity (r$_1$) was extracted from the slope of the plot of 1/T$_1$ vs. contrast agent concentration.

## Glucoverdazyl stability measurements

The EPR was tuned to a 5 mM sample of glucoverdazyl or TEMPO in PBS prior to any stability measurements. Solutions of glucoverdazyl or TEMPO were prepared (20 mM in mouse serum or 5 mM in a 4 mM sodium ascorbate buffer (pH 7.4)). A single spectrum was acquired and the peak height of the most intense peak for either compound was locked. EPR scans were then acquired every 5 s for 2 h (mouse serum) or 1.5 hr (ascorbate) to measure percent change in the activity. Storability was measured using a 5 mM solution of glucoverdazyl in water left in a fume hood exposed to light, or wrapped in tinfoil and left in a dark freezer at −20°C. Periodically, these solutions were sampled and measured by the EPR after it was tuned using a freshly prepared 5 mM sample of glucoverdazyl.

## MRI animal studies

Mice were housed under 12 h light/dark cycle, and ambient temperature of 20-24 °C and 45 to 65% humidity. All mice were provided water and fed $ad$ $libitum$ with Rodent Laboratory Chow. Mice were anesthetized with isoflurane, placed on a heated cradle and inserted into the MRI. Consecutive $T_1$-weighted RARE images as previously described above were acquired prior to and every 2.5 min thereafter for 60 min after contrast agent injection. $T_1$-weighted Imaging: slice thickness of 1 mm, FOV of 50 × 50 mm, averages = 3, matrix size = 96 × 96, TE$_{eff}$ = 11 ms, echo spacing = 7 ms, TR = 720 ms, and acquisition time of 2 min 16 s. In all cases 3 mmol/kg of contrast agent was injected $intravenously$ through a tail vein catheter, which was flushed with saline to ensure full dose of contrast agent was received.

## MRI data processing and analysis

**Intensity-over-time curves.** SUR files were used to avoid to remove any automatic gain functions associated with the MRI. All scans and slices were normalized to water-filled fiducial marker placed alongside the mice during all scans. A MATLAB routine was used to draw slice-wise regions-of-interest (ROIs) at each scan time point to generate voxel-wise intensity over time data, presented as the mean intensity of the total ROI for each timepoint normalized to 100% for the lowest intensity voxel for the first scan. A graphical representation of the following workflow can be seen in Supplementary Fig. S1.

**Renal decay time constant values and image maps.** Using the voxel-wise $T_1$-weighted intensity values for each imaging time point during the ~45 min scan, a MATLAB script extracted RDTC (k), as follows: A linear regression was fit to the natural logarithm ($ln$) of the intensity values from $t$ = 2.5 to 40 min, and the slope of this linear regression yielded the RDTC, k, $per$ voxel. The k map was then overlaid on an anatomical reference image to yield the image-based maps of kidney function. An ROI was drawn over each kidney, or over the renal cortex

and the medulla of each kidney. RDTC values shown in graphs are the mean of voxel-wise k values within an ROI.

**Area-under-the-curve values and image maps.** A baseline correction was applied to each of the voxel-wise intensity-over-time curves by subtracting the lowest voxel value, setting the baseline to 0 across all time points. The integral of this curve for each voxel was acquired using the trapezoids function to generate an area-under-the-curve (AUC) value from $t = 0$ min to $t = 50$ min. AUC values shown in graphs the mean of *per* voxel k values derived from each mouse. Maps of AUC were overlaid on top of the image acquired at $t = 0$ min.

### Glucoverdazyl tissue localization by DCE-MRI
Glucoverdazyl-enhanced scans were acquired from healthy (9) male BALB/c mice (8 weeks old) as described in the *MRI animal studies* section, and normalized intensity-over-time curves were acquired as described in the *MRI data processing and analysis* section, with ROIs drawn for kidney, liver, bladder, and muscle tissues.

**Serum creatinine measurements.** Blood was drawn from the saphenous vein at day 0 and at each post-injury time point prior to glucoverdazyl-enhanced MRI. Blood was centrifuged for 10 min (room temperature, 900 xg) and the serum was collected from the fractionated sample. Samples were stored at −80 °C until use. Serum creatinine (SCr) was determined by quantitative HPLC (Agilent 1260 Infinity with diode array equipped with a 2.1 mm × 50 mm, 5 μm particle size Agilent Zorbax 300-SCX column) against a creatinine standard curve by modifying a previously reported method[83]. Briefly, creatinine was dissolved in HPLC mobile phase (15 mM sodium acetate buffer at pH 4.2 with 4% methanol and 1% acetonitrile (AcN) and serially diluted to create a creatinine standard curve from 0 μM to 12.5 μM through integration of the produced HPLC peak at 234 nm (Bruker HyStar PP)). Standards and samples were acquired at a flow rate of 0.5 mL/min with an isocratic mobile phase. Creatinine was extracted from mouse serum samples after thawing by precipitating proteins with a 4:1 ratio of AcN and 0.5% acetic acid to serum. Samples were vortexed, then left at −20 °C for 30 min to allow complete precipitation. Samples were centrifuged at 12,000 xg (10 min, 4 °C), and the supernatant transferred to a new tube. Tubes were dried to remove acidified AcN through heated vacuum centrifugation for 45 min at 50 °C. The pellet was resuspended in 60 μl of mobile phase and samples were quantified by HPLC, integrating the matched-time elution peak seen in the standard curve at 234 nm.

### Animal models of kidney disease
Glucoverdazyl-enhanced scans for all mice in both UUO and FAN disease models were acquired as described in the *MRI animal studies* section, and normalized intensity-over-time curves were acquired as described in the *MRI data processing and analysis* section, with ROIs drawn for kidney tissue. All studies were initiated when animals were 9–10 weeks of age. All animals were housed in conventional breeding cages, fed standard rodent chow and received water *ad libitum*, and were housed in a facility with a 12-h light cycle.

### Unilateral ureter obstruction
The unilateral ureter obstruction (UUO) mouse model of acute kidney injury (AKI) was performed in C57/Bl6 mice as was previously reported in the literature[84]. Briefly, 10 female C57/Bl6 (8 weeks old) mice were sorted in to groups of 5 for either the sham procedure or UUO procedure. Mice were imaged with glucoverdazyl contrast immediately prior to surgery (day 0). Mice were anesthetized by constant isoflurane inhalation. The left kidney of the mouse was accessed laparoscopically and the left ureter was either touched gently with a surgical instrument (sham group) or tied shut with a suture (UUO group). The wound was sutured closed and mice were

imaged by glucoverdazyl contrast MRI 3-days and 7-days post injury. On day 7, mice were sacrificed by cervical dislocation. The kidneys were removed and fixed in paraformaldehyde, after which they were sectioned and stained with PAS, with images acquired using a slide scanner.

### Folic acid-induced nephropathy (glucoverdazyl contrast MRI)
The folic acid-induced nephropathy (FAN) model of AKI-to-chronic kidney disease (CKD) injury was performed in BALB/c mice following a modified procedure previously reported in the literature[85–89]. We observed a high mortality rate at doses of 250 mg/kg of folic acid (FA) administered to CD1 and C57/Bl6 mice. BALB/c mice have been shown to be more resistant to obstruction-mediated injuries and to more reliably generate CKD[69,90]. Five male BALB/c mice (8 weeks old) were imaged by glucoverdazyl-enhanced MRI at day 0. Immediately following scans, mice were injected *intraperitoneally* with 125 mg/kg FA in a 0.3 M sodium bicarbonate solution. Daily *subcutaneous* fluid support was necessary for the first 5 days following FA injection. By day 7, mice were stabilized and housed normally without fluid assistance. Mice were re-imaged by glucoverdazyl-enhanced MRI 15-days and 30-days post injury. On day 30, mice were sacrificed by cervical dislocation. The kidneys were removed and fixed in paraformaldehyde, then sectioned and stained with PAS. Images were acquired using a slide scanner. A group of mice underwent the same disease induction without any glucoverdazyl-enhanced MRI and were sacrificed at day 15 to obtain histology for this time point.

### Folic acid-induced nephropathy (transdermal fluorescence)
A parallel group of 6 male BALB/c mice (8 weeks old) had kidney disease induced identically to those in the contrast MRI group, except that transdermal fluorescence measurements, as previously described in the literature, were performed instead of MRI[78–82]. Briefly, prior to any data acquisition, hair was removed from the right dorsolateral aspect of the mice. The next day, a transdermal fluorescence monitor (MediBeacon, Inc.) was affixed to the shaved area by a proprietary windowed adhesive patch. The battery was connected to the transdermal monitor and a 5 min baseline was established. A solution of FITC-sinistrin (150 μl, 0.2 mg/kg) was injected *intravenously* through the tail vein, and data was collected for 55 min. Day 0 data collection immediately preceded intraperitoneal FA injection, and was repeated 15-days and 30-days post injury, at which point mice were sacrificed by cervical dislocation.

### Euthanasia
All animals were euthanized by cardiac puncture under isoflurane anesthesia.

### Conversion from RDTC to glomerular filtration rates
Data acquired from transdermal fluorescence was analyzed by proprietary software (MediBeacon, Inc.) to generate RDTC and glomerular filtration rate (GFR) values based pharmacokinetic model fitting. We generated GFR and RDTC data using this software with a one-phase decay model of the raw data with no corrections. MediBeacon uses a previously-determined factor that can directly convert RDTC to GFR, based on mouse data they generated and compared to measured GFR through traditional methods[78–82]. Given the consistency of our RDTCs in our 14 measured healthy BALB/c mice, we derived a conversion factor for glucoverdazyl RDTC to GFR based on the average GFR value of the 6 healthy male BALB/c mice measured through transdermal fluorescence. By comparing the difference between our pooled RDTC compared to the transdermal-derived RDTC to normalize the average GFR value, the conversion factor was derived. This conversion was applied to the mean RDTC value presented in the data to generate a GFR comparison between the two methods.

## Statistics

Statistical analyses for experiments are reported in the corresponding methods section. Normality was assumed where appropriate for all data sets. Prior to ANOVA, Levene's test was use to confirm equal variance, and visual quantile-quantile plot analysis was used to confirm homoscedasticity.

## Reporting summary

Further information on research design is available in the Nature Portfolio Reporting Summary linked to this article.

## Data availability

All data can be found in the main manuscript or the SI, quantified source data associated with the current study are available at https://doi.org/10.6084/m9.figshare.23538426. Raw animal images can be obtained from the corresponding author upon request due to large file size constraints.

## Code availability

MATLAB code for generating quantitative RDTC and AUC data, as well as their associated functional maps, is available at https://github.com/ncalvert994/Glucoverdazyl.

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

## Acknowledgements

The work reported was supported by funding from the Canadian Institutes for Health Research (PJT-180355). N.D.C. and A.K. acknowledge graduate support from the Ontario Graduate Scholarship program. The authors would like to thank F. Mercury for support during MRI acquisitions, Dr. XiaoLing Zhao for the many histological samples she prepared and scanned, as well as Dr. Christopher Boddy and André R. Paquette for use of their HPLC system for serum creatinine measurement.

## Author contributions

N.D.C. and M.S. performed all synthesis and characterization of chemical compounds with support from A.A.T.. N.D.C. performed all in vitro physicochemical and cell-based experiments. P.P. performed NMR relaxivity and mechanism of contrast enhancement studies. N.D.C. and A.K. performed animal studies, including model development, implementation, and imaging. N.D.C. wrote the MATLAB code for imaging analysis with support from G.M., and performed all analyses. N.D.C. performed statistical analyses. N.D.C., A.J.S., and A.K. conceived of experiments with support from D.B. and N.S.. N.D.C. and A.J.S. co-wrote the paper, and all authors edited the work and provided input on response to reviewers.

## Competing interests

The authors declare no competing interests.
