## [Peer Review File · Nature Communications]

Direct Mapping of Kidney Function By DCE-MRI Urography Using a Tetrazinanone Organic Radical Contrast AgentREVIEWER COMMENTS

Reviewer #1 (Remarks to the Author):

This work by Calvert et al. reports developing and applying a glucose-modified verdazyl, glucoverdazyl. Glucoverdazyl is an organic radical contrast agent with a strong EPR signal derived from the radical nitrogen-centered π -orbitals on a tetrazinanone ring. The hypothesis is that the stable glucoverdazyl can be used as a gadolinium-free MRI contrast agent (CAs). Due to the fast clearance and simple renal pharmacokinetics can be used to accurately determine glomerular filtration rates in healthy and disease mouse models. Similar to what is done already with Gd-based CAs. This study includes in vitro studies with phantoms to validate the MR contrast (r_1 relaxivity), cell work to determine cytotoxicity, and uptake with hRPT cells. It also describes the in vivo application of glucoverdazyl (determination of GFR) in healthy and unilateral ureteral obstruction (UUO) and folic acid nephropathy (FAN) mouse models.

The work is generally well presented, and the studies follow logically from the initial premises. This work aims at the in vivo determination of GFR using a stable glucoverdazyl organic radical by MRI. It is very relevant and of broad interest. It is also quite cross-disciplinary.

Nevertheless, the work sometimes lacks novelty, focus, and cross-validation studies, particularly in preclinical translation.

Key results:

- Organic synthesis of a stable MRI glucoverdazyl ORCA.
- In vitro validation of MR contrast detection, reduction kinetics, and cytotoxicity.
- In vivo application of glucoverdazyl in healthy and kidney dysfunctional mouse models.
- Determination of accurate GFR from fast clearance of glucoverdazyl with unparalleled spatial resolution.

The work could be considered for publication in Nature Communications; however, significant revisions would be recommended before publication.

Comments:

Study design and methodology:

- Generally well-designed studies and excellent selection of the methodology.
- The authors showed that glucoverdazyl is quite resistant to reduction with ascorbate. It is not surprising because the nitril group is not stabilized upon reduction. However, upon oxidation with superoxide, xanthine oxidase, peroxides, and metal-based oxidation through Fenton reactions, the question remains. In principle, the radical should quickly transfer to some of these strong acceptors. Authors should consider including these experiments to test for stability under redox states.
- The r_1 relaxivity was determined at 3T, showing a very small r_1 . At concentrations below 1mM, the albumins will show higher R_1 s than the glucoverdazyl. Authors should study the binding of glucoverdazyl (0.1mM) to human serum albumin at physiological levels.
- The authors should elaborate further on the mechanism of paramagnetic relaxation with this system. Is the effect related to second-sphere relaxation? The fact that T_2 does not change at 3T implies that the Curie effect might not play the most prominent role in the proton relaxation effect or that the waters might be placed at the slower exchange regime. A study with 3 or 4 different temperatures could partially elucidate the mechanism. If the authors have access to a high-field NMR spectrometer (4.7-11.6 T), comparing the T_1 and T_2 contributions could be interesting.
- The rationale for selecting the cells for cytotoxicity and ORCAs uptake presented here is unclear. Authors should consider including cell lines with more GLUT and MCT expression to check for the uptake of glucoverdazyl and better support the conclusions. This should be validated with western blots and flow cytometry.
- The preclinical work has been performed in different mouse backgrounds: BALB/c and C57/Bl6 mice. The control experiments were performed in BALB/c mice. A justification for the use of the mice is required. How comparable are the models so that conclusions can be well supported? For instance, $n=9$ in this experiment (see biostatistical justification below).
- Compared to standard renal excreted Gd-based CAs (Gadavist, Magnevist, Dotarem), how accurate is the GFR determination with glucoverdazyl? Also, authors should compare the performance of

gluoverdazyl compared to verdazyl. What is the real benefit of using the linear gluconate compared to the other similar ORCAs (i.e., cytotoxicity is also meager with trityls and TEMPO derivatives, so what is special about this organic radical compared to the other stable ORCAs?)?

- What is the effect of the pH on the EPR and MR signal (r_1 relaxivity)?

- Figure 3, it is unclear how the authors calculated the RDTC maps. What can we conclude from the imaging?

- The justification for the mismatch between the GFR determined by transdermal fluorescence and dynamic contrast-enhanced magnetic resonance imaging (DCE-MRI) in folic acid nephropathic (FAN) mice (Figure 4) is not convincing. One should see a clear difference on days 0, 15, and 30, not only on days 0 and 15. What happened on day 30? Transdermal fluorescence seems to perform better in identifying those differences, even though statistically, the SD is huge. Can the authors elaborate on this?

- The calculation of GFR and RDTC is not well described. It seems that the authors calculated the first decay with a pseudo-first order inverse kinetic model, but the clearance combines a second-order kinetic value originating from the accumulation of probes in the diseased models. How was this accounted for? A linear fitting will not serve this purpose, I am afraid. Could this explain the mismatch between transdermal fluorescence and dynamic contrast-enhanced magnetic resonance imaging?

- Authors should tone down some of the statements in the text. For instance:

o "Regardless of the Gd-based contrast agent used, ~20% of the injected dose is deposited irreversibly in the body, including the bone and brain." Rather, 1-2% of patients were injected continuously with labile Gd-complexes for several days. For stable Gd-complexes, Gd is sometimes found in the brain at ppm levels in the form of phosphonates (Radiology: Cardiothoracic Imaging 2019; 1(3):e190104 and J. Magn. Reson. Imaging 2009;30:1259–1267)

Conclusions:

- The data do not support some conclusions/findings. For example: "...allowing the acquisition of quantitative and qualitative kidney functional information." The imaging provided is rather qualitative, not quantitative. The calculated kinetic values are relative to the dose, perfusion, and proton-base catalyzed events. This is a strong declaration that is not supported at all by the data provided.

Scholarly presentation:

- The presentation of the data should be improved in most figures. The letters are too small in the labels and legends from the graphs and plots from the central figures (practically unreadable).

- The resolution of figures S12-S14 is low.

Appropriate use of statistics and treatment of uncertainties:

- The biostatistics methods and analysis are not well justified. Authors should comment on the sample sizes chosen to ensure adequate power for the predicted effect size. So, how exactly the authors determined $n=9$ for the in vivo experiments?

- The authors should provide more information about the following:

o the expression of the size effect.

o comparison of the groups, and matched conditions. For instance, fasted animals, ad libitum, glucose levels, weight, etc.

o Assumption for the normality and paired data.

o the appropriateness of the statistical test selected for this work.

Thank you!

Reviewer #2 (Remarks to the Author):

Summary

The authors demonstrate use of glucoverdazyl for the assessment of GFR by MRI in mice with acute kidney injury (surgical or medical) and chronic kidney disease (at 15 and 30 days).

Strengths:

1. This study tests a potentially novel verdazyl for use at MRI as a novel contrast agent for assessing GFR. In current state, the methods used for assessing GFR have some limitations.

Key Weaknesses:

1. The rationale and background for the method deserve additional support and argumentation in comparison to current state with respect to clinical applicability and usability of spatial information, and to the pragmatic application of MRI-based methods for assessing GFR.
2. In current state, DCE-MRI using existing molecules is feasible and safe, even in patients with kidney disease.

Specific comments

Title

1. OK

Abstract

1. Sentence 2: Revisions to eGFR formulae (implemented in the last 2-3 years) and now used in clinical practice have omitted race from their designations.
2. Sentence 3: The clinical value of spatial information within a kidney as it relates to kidney function should be alluded or specified.
3. In current practice, kidney function assessment is by scintigraphy. The rationale for replacing it with DCE-MRI should be stated.

Introduction

1. Line 46: Please specify the interventions that are possible once CKD is diagnosed to prevent progression.
2. Line 55: It is recognized that eGFR is biased relative to true GFR and is especially biased in formulas that were derived from biased populations. More accurate tests that incorporate cystatin C are available.
3. Line 58: It is an overstatement to say it is "imperative", but it certainly would bring value to have more accurate and easily-obtained methods. MRI-based methods of measuring GFR probably are not scalable to the general population or usable for screening; therefore, they do not probably qualify as easily obtained. In other words, even if it is perfectly accurate, from a practical standpoint, obtaining MRI in all patients who otherwise would have eGFR measured is not feasible. Essentially all patients who come to the ED or are inpatients undergo eGFR testing. In addition, many outpatients with risk factors also undergo testing. It is extremely common. The commonality is problematic because any test that isn't easy, cheap, and broadly accessible isn't going to work as a substitute in the vast majority of patients. As a partial analogue to this, cystatin C is much more accurate than serum creatinine, but lack of availability and familiarity has limited uptake. Translate that same idea to MRI and it is orders of magnitude more challenging / infeasible.
4. Line 61: It is stated that spatial information is "important information" for patients with CKD. This theme is repeated in the manuscript without explanation for how it would be helpful. There are not current clinical paradigms that make significant use of or would require this. Please explain how within-kidney spatial information is helpful or will be helpful in the future.
5. Line 68: When describing the potential limitations of scintigraphy, please include both DTPA and MAG-3 (MAG-3 being more accurate than DTPA in patients with CKD or AKI). In the current description, only DTPA is included. Also, rather than a just a listing of potential weaknesses of scintigraphy, please provide specific numeric accuracy and precision data for both scintigraphic

methods (with reference standards) in patients with kidney disease so the reader knows what the current state is that MRI is competing against.

6. Line 85: The greatest challenge for DCE-MRI in evaluation of GFR is not perceived risk of gadolinium (although I agree that is a barrier). Even if that issue is off the table, the greatest challenge is pragmatic application of an expensive and time-consuming test as a replacement for pragmatic alternatives.

7. Line 90: Please include in your discussion of NSF the differentiation of various types of GBCM. Some agents (gadobenate, macrocyclics) have extremely low risk of NSF, whereas other agents are high risk.

8. Line 96: It is not accurate to say that 20% of the dose is deposited “Irreversibly” in tissues. The fraction of retained gadolinium is much smaller (<6%) than that, the fraction that is retained is slowly cleared over months or years, and the likelihood of retention is highly dependent on the agent (macrocyclic agents are retained a further order of magnitude less than linear agents).

9. Line 131: (re: “toxicological concerns”): Modern GBCAs are incredibly safe and extensively tested. For modern agents, there is a tiny theoretical risk of NSF (0 to 0.07%), a rare risk of severe contrast reaction (less than 1 in 100,000), and common but tiny amounts of gadolinium retention that are of unclear and doubtful significance in most patients. New agents will have a very high bar to pass to replace them because of the extensive testing that would be involved. During that extensive testing, it is extremely likely that rare side effects will emerge just as they did for gadolinium-based media. In the present state, contrast-enhanced MR based kidney perfusion and quantification can be performed with modern GBCAs. The reason it isn't done is not truly because of risk of NSF or retention. It is because there are more practical ways of acquiring the information, and there is not a present dominant clinical need for spatial kidney function data.

Results and Discussion

1. Defer to chemists for evaluation of the synthesis and chemistry.

2. Line 205-210 & Figure 2: Enhancement is observed in muscle and liver, even up to 40 mins post injection.

3. Line 343: Why did the fibrosis decrease on day 30? This seems very unusual to have rapid decline in fibrosis.

4. Line 374: What is the reference standard on which transdermal fluorescence monitoring was validated? Further, has it been validated at day 0, day 15, and day 30?

5. Line 398: Please do not speculate about trends that are not statistically significant.

Conclusion

1. There is substantial reference in the manuscript about the use of the technique for spatial mapping of kidney dysfunction. However, the methods and results presented to not assess kidney function differently within spatial regions of individual kidneys.

References

1. Please include modern references 2021-2023 that discuss use of alternative methods of kidney function assessment by blood tests that do not use race and are now widely used in clinical care.

Figures and Tables

1. Good

Reviewer #3 (Remarks to the Author):

The authors describe the use of a novel organic radical based MRI contrast agent as a way to examine kidney function using a mouse model. The work is significant because of the established toxicity of standard gadolinium contrast agents in individuals with impaired kidney function. Important results that form part of this paper include the establishment of the relative non-toxicity of this class of contrast agents, their relatively long lifetime in vivo, and their efficacy as contrast agents. The methodology regarding the synthesis and characterization of the contrast agent is sound; the provided data establishes the identity of the material and there is certainly sufficient information to reproduce the synthesis. Establishing purity of stable free radicals is a little more challenging than many other organic compounds because they do not give sharp NMR spectra. The authors provide HPLC data that supports the purity of the samples, but additional information such as elemental analysis or UV-spectra (which can be compared with published data for the previously reported synthesis) would provide an even more solid foundation.

The reported cytotoxicity and animal studies appear to support the conclusions regarding toxicity and efficacy as a contrast agent; however, as a chemist less familiar with such biomedical studies I cannot comment on this aspect of the paper in more detail.

In general though, the significance of this study, in terms of breaking new ground for MRI contrast agents, makes it well worth publication in Nature Communications

REVIEWER COMMENTS

Reviewer #1 (Remarks to the Author):

- The authors showed that glucoverdazyl is quite resistant to reduction with ascorbate. It is not surprising because the nitril group is not stabilized upon reduction. However, upon oxidation with superoxide, xanthine oxidase, peroxides, and metal-based oxidation through Fenton reactions, the question remains. In principle, the radical should quickly transfer to some of these strong acceptors. Authors should consider including these experiments to test for stability under redox states.

We have extended our evaluation of glucoverdazyl stability under all of these conditions. The simplest interaction to evaluate was the effects of known oxidizers and reducers: hydrogen peroxide and reduced glutathione, respectively. As shown in Fig. 1, the addition of either of these in excess to a solution of glucoverdazyl showed no change in radical activity as monitored by EPR.

Response Figure 1. Glucoverdazyl stability in the presence of known oxidizers and reducers. Twenty millimolar of either hydrogen peroxide (H₂O₂) or reduced glutathione (GSH) were added to a solution of 5 mM glucoverdazyl in phosphate-buffered saline (PBS). These solutions were monitored by an EPR that was previously tuned to a 5 mM sample of glucoverdazyl in PBS. Solutions were monitored for 90 min to determine if any loss of radical activity occurred by either oxidation or reduction of glucoverdazyl.

We then tested how glucoverdazyl might affect Fenton chemistry by looking at the classical example of FeCl₂ oxidation to FeCl₃ by H₂O₂. In previous work by Shi *et al.* (<https://doi.org/10.1016/j.bbrc.2016.12.174>), authors were able to show that oxidation of Fe (II) citrate to Fe (III) citrate was significantly inhibited in the presence of TEMPO. Interestingly, the authors showed that in the presence of a common spin trap (DMPO), the DMPO signal was reduced when TEMPO was added. However, TEMPO signal was unchanged. The authors attributed this effect to a free-radical scavenging effect of

TEMPO with H₂O₂, significantly reducing Fe(II) to Fe(III) oxidation while also preserving the EPR intensity of TEMPO. We attempted to mimic this experiment using glucoverdazyl in place of TEMPO. To a 20 mM acetate buffer containing 1 mM of FeCl₂, 10 mM DMPO, mM of either TEMPO or glucoverdazyl was added 10 mM of H₂O₂. EPR spectra were recorded immediately upon the addition of H₂O₂ and again 1 hr later. Response Fig. 2A shows the evolution of hydroxyl radical species as trapped by DMPO. The trapped product signal was lost after 60 min (Response Fig. 2D) due to the very short lifetime of DMPO trapped radicals (<https://doi.org/10.1016/j.aca.2004.02.020>). Response Fig. 2B and 2E, as well as 2C and 2F, show no change in EPR maxima in solutions containing either TEMPO or glucoverdazyl, respectively. By subtracting the 0 min and 60 min TEMPO spectra from one another, a very faint DMPO signal appears that is in agreement Shi and coworkers' data not shown). No discernible DMPO signal was observe upon subtraction of the glucoverdazyl EPR due to the complexity of the glucoverdazyl spectrum masking any DMPO signal. While it was evident that neither glucoverdazyl or TEMPO were losing radical activity as a result of the presence of the hydroxyl radical, we knew from the work of Shi *et al.* that there may be a free radical scavenging or catalytic effect to account for. To determine changes in the degree of Fe(II) to Fe(III) oxidation, we did a simple absorbance assay looking at the change in Fe(III) absorbance at 330 nm (corrected for any absorbance by our radical molecules, Response Fig. 2H). In agreement with Shi *et al.*, there was almost no turnover of Fe(III) in the presence of TEMPO and H₂O₂. With glucoverdazyl, however, we saw a two-fold enhancement in oxidation. While a detailed explanation is outside the scope of this paper, we have proposed a possible explanation (Response Scheme 1). The oxidation of glucoverdazyl in it's non radical form proceeds through Fe(III) mediated reduction (ferricyanide) to Fe(II) (ferrocyanide). We believe that in the presence of Fe(II) and H₂O₂, there is oxidation of the verdazyl radical to verdazylum as shown by Gilroy *et al.* in 2007 (<https://doi.org/10.1021/ol702163a>) as well as a number of other publications through cyclic voltammetry. This oxidation is likely facilitated by the hydroxyl radical. The verdazylum is then reduced by Fe(II) back to the verdazyl, generating Fe(III) as the oxidation product. This would create a catalytic scenario where we would not see a change in EPR activity, but we would then see an increase in Fe(III) production. We therefore believe that while there is interaction between Fenton chemistry and glucoverdazyl radical, there is no net effect on the amount of paramagnetic species.

Response Figure 2. Interactions of TEMPO and glucoverdazyl with Fenton chemistry by H₂O₂ mediated oxidation of FeCl₂ to FeCl₃. EPR spectra of 10 mM DMPO alone or with 5 mM of either TEMPO or glucoverdazyl before (A, B, C) and 1 hr after (D, E, F) the addition of 10 mM H₂O₂ to initiate oxidation of FeCl₂ to FeCl₃. G) The change in maximum EPR signal before and 1 hr after oxidation reaction. H) Conversion of Fe(II) to Fe(III) determined by change in absorbance at 330 nm.

Response Scheme 1. Verdazyl catalyzed oxidation of Fe(II) to Fe(III).

Finally, we also evaluated the stability of glucoverdazyl in the presence of superoxide produced through the xanthine/xanthine oxidase (X/XO) system. We see no change in radical activity for both TEMPO and glucoverdazyl by EPR in the presence of xanthine (X), xanthine oxidase (XO), and both xanthine/xanthine oxidase (X/XO) (Response Fig. 3). In anticipation of any catalytic reactions between these radicals and superoxide, though none were evaluated in the literature, we again performed a colorimetric assay to evaluate the change in conversion of xanthine to uric acid through xanthine oxidase. However, no change was observed. Somewhat surprisingly, given the similarity in structure between the core of glucoverdazyl and xanthine, there was no reaction between glucoverdazyl and XO. Therefore, glucoverdazyl appears to be stable against superoxide, especially in concentrations that would be encountered *in vivo*.

Response Figure 3. Interactions of TEMPO and glucoverdazyl with superoxide produced by xanthine oxidase. EPR spectra of 1.25 mM TEMPO containing 10 mM xanthine (X, Fig. 3A), 0.4 U xanthine oxidase (XO, Fig. 3B), or both (X/XO, Fig. 3C). Figures 3D, E, and F are the same but with 1.25 mM glucoverdazyl. The change in EPR maxima compared to the tuned radical is shown in Fig. 3G. Relative uric acid production as a result of the X/XO reaction is shown in Fig. 3H by change in absorbance at 293 nm.

We have included the following additions to the manuscript to indicate the results of these experiments, and have included the figures in the Supporting Information as indicated in the text below:

“Glucoverdazyl stability was also evaluated in the presence of hydrogen peroxide and glutathione (Supplemental Fig. S10), the Fenton cycle (Supplemental Fig. S11), and superoxide derived from the xanthine-xanthine oxidase reaction (Supplemental Fig. S12). In all cases, glucoverdazyl maintained its radical character.”

The following text was added to the experimental section:

“The EPR was tuned to a sample of glucoverdazyl or TEMPO in PBS prior to any of the stability measurements. Once tuned, solutions of glucoverdazyl or TEMPO were prepared (20 mM in mouse serum or 5 mM in either 4 mM sodium ascorbate buffer (pH 7.4), 10 mM glutathione

(GSH), or 10 mM hydrogen peroxide (H_2O_2)). A single spectrum was acquired and the peak height of the most intense peak for either compound was locked. EPR scans were then acquired every 5 s for 2 hr (mouse serum) or 1.5 hr (ascorbate, H_2O_2 , and glutathione) to measure percent change in the activity. For stability measurements of glucoverdazyl in water, a 5 mM sample was prepared and left in a fume hood exposed to light, or wrapped in tinfoil and left in a dark freezer at $-20^\circ C$. Periodically, these solutions were sampled and measured by the EPR after it was tuned using a freshly prepared 5 mM sample of glucoverdazyl.

Glucoverdazyl interaction with hydroxyl radicals generated through Fenton reactions was evaluated through $FeCl_2$ to $FeCl_3$ oxidation catalyzed by H_2O_2 . A 20 mM ascorbate buffer (pH = 4.5) containing 1 mM $FeCl_2$ and 10 mM 5,5-dimethyl-1-pyrroline-N-oxide (DMPO). This solution was sparged with N_2 for 30 min and aliquoted into vials containing glucoverdazyl or TEMPO to make a 5 mM concentration of either radical. Immediately prior to EPR scan, 10 mM H_2O_2 was added to the solution. After mixing, the EPR activity was measured, with measurement repeated after 1 hr. Aliquots of the 1 hr solution were taken and absorbance was measured at 330 nm to determine the amount of $FeCl_3$ produced by $FeCl_2$ oxidation by H_2O_2 . Absorbance of TEMPO and glucoverdazyl without $Fe(II)$ were subtracted from the oxidized solutions of each to account for inherent absorbance of these molecules.

Glucoverdazyl interaction with superoxide was evaluated by xanthine (X)/xanthine oxidase (XO) reaction. Solutions of either TEMPO or glucoverdazyl (1.25 mM) in PBS were made containing either 10 mM X, 0.4 U XO, or both. These solutions were incubated under cell culture conditions for 30 min before EPR activity of the solution was measured. Aliquots of the solution were taken and absorbance was measured at 293 nm to determine the production of uric acid, the product of the X/XO reaction. Absorbance of TEMPO and glucoverdazyl without X, XO, or X/XO were subtracted from the oxidized solutions of each to account for inherent absorbance of these molecules.”

- The r_1 relaxivity was determined at 3T, showing a very small r_1 . At concentrations below 1mM, the albumins will show higher R_1 s than the glucoverdazyl. Authors should study the binding of glucoverdazyl (0.1mM) to human serum albumin at physiological levels.

We sought to investigate the binding of glucoverdazyl to human serum albumin (HSA) at physiological levels, as well as probe the HSA binding sites putatively interacting with glucoverdazyl adapted from procedures from Caravan *et al.* (2002, 10.1021/ja017168k). Glucoverdazyl binding to HSA at physiological concentration (4.5% w/v) was evaluated after incubation for 30 min at $37^\circ C$ and 5% CO_2 under gentle shaking (Response Fig. 4A). The free and bound glucoverdazyl were separated by Centrifree[®] Ultrafiltration filters. The free fractions were quantified by HPLC against a previously determined standard curve. There was almost no glucoverdazyl binding to HSA at concentration below 1 mM. Even at higher concentrations (3 mM) there is only a very small amount of binding which is unsurprising for compounds in very high concentration. The results of the site binding assay were similar to those seen by HPLC; almost no binding up until the 1 mM mark, with a small amount of binding (consistent with amounts seen by HPLC) in both site I and site II (Response Fig. 4B), which is likely due to high concentration non-specific binding.

Response Figure 4. Glucoverdazyl binding to human serum albumin (HSA). A) Glucoverdazyl fraction bound to a 4.5% w/v solution of HSA in PBS as determined by HPLC following ultrafiltration of a glucoverdazyl/HSA solution, separating bound from free fractions. B) Occupation of HSA binding site I or II by glucoverdazyl as determined fluorescently by displacement of site I or site II specific dansylamide or dansylglycine, respectively.

We next sought to investigate the effect of the presence of HSA on T_1 -shortening by glucoverdazyl. We measured the relaxation rate of a 0.1 mM and 1.0 mM glucoverdazyl solution in different amounts of HSA, adapted from experiments performed by Caravan *et al.* in 2002 (10.1021/ja017168k). Similar to what was observed in the HSA binding experiments, there was very little change in glucoverdazyl relaxation rate at HSA concentrations at and below a physiological level (Response Fig. 5). At higher concentrations of HSA, there was quite a large increase in relaxation rate.

Figure 5. Relaxation rate of glucoverdazyl in varying concentrations of human serum albumin (HSA). Relaxivity maps of NMR tubes filled with glucoverdazyl (0.1 mM or 1 mM) in either PBS or varying concentrations of HSA (numerically listed in the image as w/v) as determined by MRI at 3 T. A graphical representation below of relaxation enhancement, ϵ^* ($\frac{1}{T_{1HSA}} / \frac{1}{T_{1PBS}}$) in the presence of HSA.

We have included the following additions to the manuscript to indicate the results of these experiments, and have included the figures in the Supplemental Information (Supplemental Figure 14) as indicated in the text below:

“The binding of glucoverdazyl to human serum albumin (HSA) at physiological levels was investigated *in vitro* in order to better define the putative mechanism of MRI contrast enhancement *in vivo*. The fractional binding of glucoverdazyl was evaluated (Supplemental Fig. 14A), demonstrating negligible glucoverdazyl binding to HSA at concentration below 1 mM and an increase in binding with [HSA] > 1 mM driven by mass action. These results were

recapitulated by the investigation of fluorophore displacement from albumin binding site I and site II (Supplemental Fig. 14B). Again, no binding at either site was noted below 1 mM HSA, with a slight increase in fluorophore displacement in both sites I and II suggestive of non-specific interactions driven by mass action. Finally, we evaluated the effect of the presence of HSA on T_1 -shortening by glucoverdazyl (Supplemental Fig. 14C). The relaxation rate of a 0.1 mM and 1.0 mM glucoverdazyl solution was measured in the presence of HSA varying from 0% to 22.5% w/v in PBS. Similarly, to what was observed in the HSA binding experiments, there was very little change in glucoverdazyl relaxation rate at HSA concentrations at and below physiological levels (i.e. [HSA] < 1 mM), before mass action-induced non-specific binding occurred. These data suggest that glucoverdazyl will not interact with serum albumin once injected *intravenously*.”

The following text has also been added to the experimental section:

“For relaxivity measurements, the same imaging phantom was used with contrast agent concentrations of 1 to 3 mM, 1 mM or 0.1 mM contrast agent in 0% to 22.5% HAS in PBS, or 3 mM contrast agent in PBS of pH 3, 7, or 11. Contrast agent concentration following imaging was verified by electron paramagnetic spectroscopy. To measure the longitudinal relaxation rate (R_1), an inversion recovery RARE sequence was implemented with the following parameters: Slice thickness of 5 mm, FOV of 50×50 mm, average = 1, matrix size = 96×96 , $TE_{\text{eff}} = 17$ ms, TR = 5,000 ms, TI = 50, 75, 100, 150, 200, 250, 300, 400, 600, 800, 1200, 2400, and 4800 ms, and acquisition time of 2 minutes 30 seconds *per* TI. Longitudinal relaxation rates were extracted using the mapping MATLAB 2 routine written by J. Barral, M. Etezadi-Amoli, E. Gudmundson, and N. Stikov (2009), and modified by J. Rioux (2022). The longitudinal relaxivity (r_1) was extracted from the slope of the plot of $1/T_1$ vs. contrast agent concentration.”

“Human serum albumin binding assays for both binding and site specificity were adapted from methods by Caravan *et al.*, 2002. A solution of 4.5% w/v human serum albumin (HSA) containing 3 mM glucoverdazyl was prepared. The solution was serially diluted with a 4.5% HSA solution in PBS. The dilutions were incubated under cell culture conditions for 30 min. Afterwards, free glucoverdazyl was separated from bound glucoverdazyl through ultracentrifugation separation using Centrifree® Ultrafiltration filters (2000 xg, 20 min). The unbound glucoverdazyl in the filtrate was quantified by HPLC using the protocol described in Supplemental section 3.2, after developing a glucoverdazyl HPLC standard curve in PBS. The concentration of bound glucoverdazyl was calculated by subtracting the added concentration of glucoverdazyl from the HPLC determined concentration of free glucoverdazyl.

Site specificity of glucoverdazyl binding was determined by displacement of site-specific fluorophores (dansylamide, site I; dansylglycine, site II). A solution of 4.5% w/v HSA with 3 mM glucoverdazyl was prepared containing 50 mM dansylamide or dansylglycine. These solutions were serially diluted with a 4.5% w/v HSA solution containing 50 mM of either probe but no glucoverdazyl. Another set of solution containing the glucoverdazyl dilutions in HSA but with neither probe was also prepared. These dilutions were incubated under cell culture conditions for 30 min. The dilutions were aliquoted in a black-walled 96-well plate. Fluorescence was measured (ex. 365 nm/em. 480 nm) for each well. Intensity values for glucoverdazyl without probe were subtracted from the same concentration containing probe to account for absorbance incurred by the glucoverdazyl molecule. Corrected fluorescence values were compared to the either fluorescent probe in HSA containing no glucoverdazyl to determine the degree of probe displacement from HSA caused by glucoverdazyl binding.”

- The authors should elaborate further on the mechanism of paramagnetic relaxation with this system. Is the effect related to second-sphere relaxation? The fact that T2 does not change at 3T implies that the Curie effect might not play the most prominent role in the proton relaxation effect or that the waters might be placed at the slower exchange regime. A study with 3 or 4 different temperatures could partially elucidate the mechanism. If the authors have access to a high-field NMR spectrometer (4.7-11.6 T), comparing the T1 and T2 contributions could be interesting.

Water relaxation rates ($R_1 = 1/T_1$ and $R_2 = 1/T_2$) were studied between 10°C – 80°C at 7.05 T (ν_0 (^1H) = 300.15 MHz). Both glucoverdazyl (at 3 mM) in PBS buffer and pure buffer solutions were examined in a mixture of 4:1 H₂O:D₂O to provide a field lock. In order to look at the possible mechanism of relaxation for glucoverdazyl, the corrected relaxation rate was examined, which is the difference of the relaxation rate of water in the presence of the probe and the relaxation rate of water in pure buffer ($R_{1,2}(\text{corr}) = R_{1,2}(\text{probe}) - R_{1,2}(\text{buffer})$) (Response Fig. 6A). While there was a small temperature-dependent increase in R_2 , there was no significant R_2 relaxation observed in the T₂-weighted MRI image of the probe. It is, then, possible there is a negligible contribution to relaxation *via* the Curie spin relaxation mechanism (higher R_2 at lower temperature and higher field) (*Magn. Res. Med.* **2001**, 46, 917-922). The negligible Curie contribution is unsurprising as the spin quantum number (J or S) ($S = 1/2$) for ORCAs is lower than that for GBCAs, with electronic relaxation times (T_{1e}) being much larger for ORCAs (μs to high ns vs ps or less) (*Distance Measurements in Biological Systems by EPR; Magn. Res. Med.* **2001**, 46, 917-922). This means that the inner-sphere relaxation mechanism would likely be dominated by Solomon-Bloembergen relaxation and chemical exchange (*Magn. Res. Med.* **2001**, 46, 917-922; *Chem. Rev.* **1999**, 99, 2293-2352). Given the low relaxation rates of glucoverdazyl at high field (especially R_2), it is also possible there is an outer-sphere contribution to the relaxation rate as well (*Chem. Soc. Rev.* **2006**, 35, 512-523).

Response Fig. 6. Temperature-dependence of the longitudinal (circle) and transverse (square) relaxation rates of glucoverdazyl, corrected for buffer effects, at 7.0 T

The following has been added to the manuscript, and the figure has been added to the supplemental information (Supplemental Figure 9):

“In order to look at the possible mechanism of water relaxation for glucoverdazyl, the temperature-dependence of the corrected relaxation rate was examined (Supporting Fig. 9). The small temperature-dependent increase in R_2 and lack of significant T_2 -weighted image enhancement (Fig. 1B) suggests a negligible Curie spin relaxation mechanism (*Magn. Res. Med.* **2001**, *46*, 917-922). The negligible Curie contribution is unsurprising as the spin quantum number (J or S) ($S = 1/2$) for ORCAs is lower than that for GBCAs, with electronic relaxation times (T_{1e}) being much larger for ORCAs (μ s to high ns vs ps or less) (*Distance Measurements in Biological Systems by EPR; Magn. Res. Med.* **2001**, *46*, 917-922). Inner-sphere relaxation mechanisms would likely be dominated by Solomon-Bloembergen relaxation and chemical exchange (*Magn. Res. Med.* **2001**, *46*, 917-922; *Chem. Rev.* **1999**, *99*, 2293-2352). Given the low relaxation rates of glucoverdazyl at high field (especially R_2), it is also possible there is an outer-sphere contribution to the relaxation rate as well (*Chem. Soc. Rev.* **2006**, *35*, 512-523).”

The following has been added to the experimental section:

“High field NMR relaxation measurements were performed at 7.05 T ($\nu_0(^1\text{H}) = 300.15$ MHz) on a Bruker Avance spectrometer. Samples were dissolved in PBS buffer in a mixture of 4:1 $\text{H}_2\text{O}:\text{D}_2\text{O}$. D_2O was used as a field lock. ^1H NMR experiments on pure water at high field with a high Q probe will produce significant radiation damping (*Phys. Rev.* **1954**, *95*, 8-12). So, commonly used inversion recovery experiments for determining T_1 will result in artificially smaller T_1 measurements due to small amounts of transverse magnetization produced with slightly imperfect inversion pulses. This was the case for these samples (data not shown). To alleviate this outcome, a saturation recovery experiment was used to determine T_1 . A 3 s CW saturation pulse was used on-resonance prior to the variable delay and read pulse to collect time-arrayed saturation recovery data. Ten – twelve time points were collected at each temperature. T_2 data was collected using a Car-Purcell-Meiboom-Gill (CPMG) sequence with a 20 ms total echo time and an appropriate number of loops to ensure complete loss of signal at each temperature. Eleven time points were collected at each temperature. A relaxation delay of at least $5 \cdot T_1$ was used for T_1 and T_2 measurements at each temperature. T_1 data was fit using Bruker Topspin 4.1.4 and T_2 data was fit using GNAT (*Magn. Res. Chem.* **2018**, *56*, 546-558).”

- The rationale for selecting the cells for cytotoxicity and ORCAs uptake presented here is unclear. Authors should consider including cell lines with more GLUT and MCT expression to check for the uptake of glucoverdazyl and better support the conclusions. This should be validated with western blots and flow cytometry.

Our original rationale for the use of H460 was for general cytotoxicity. Following discovery of high levels of localization to the kidney through urinary system excretion, we chose to re-evaluate cytotoxicity in hRPT cells due to substantial accumulation.

We have extended our contrast agent uptake experiments in an additional cell line, human hepatocellular carcinoma (HEPG2). HEPG2 cells have a high level of expression of GLUT1 (<https://doi.org/10.1016/j.bbamem.2007.11.015>) and MCT (<https://doi.org/10.3727%2F096504017X14902648894463>) transporters. We performed the same experiment as we did with hRPT, however this time we used glucose free media prior to the addition of glucoverdazyl to starve the cells of glucose and further drive any potential uptake (ref <https://onlinelibrary.wiley.com/doi/full/10.1002/bit.22799>)

(Response Fig. 6). We saw no significant glucoverdazyl uptake as measured by EPR in these cells.

Response Figure 6. Glucoverdazyl uptake in human hepatocarcinoma cells (HEPG2). HEPG2 cells were grown to confluency and starved for 1 hr with glucose-free media. Cells were then incubated with 10 mM glucoverdazyl in the glucose-free overnight. Cells were washed, pelleted, and measured by EPR to determine glucoverdazyl concentration internalized in the cells.

We have included the following additions to the manuscript to indicate the results of these experiments as indicated in the text below, and have added the data as panel B in Supporting Fig. 17.

“The cell uptake of glucoverdazyl by hRPT cells and glucose-starved HepG2 cells was evaluated by EPR spectroscopy, and indicated that glucoverdazyl was not taken up in any detectable manner (Supplemental Fig. S17).”

We have also added the following to the methods section regarding the experiment:

“HEPG2 cells were grown in RPMI supplemented with 10% FBS and 1% P/S until 80% confluent, at which point they were passaged. Cells were passaged three times before being seeded in to 6-well plates and grown until 80% confluent. Cell media was changed to DMEM supplemented with 10% FBS and 1% P/S, containing no glucose for 1 hr. The media was replaced after 1 hr containing 0 or 10 mM glucoverdazyl. The cells were then incubated for 24 hrs. The media was aspirated and cells were washed three times with 37°C with PBS. Cells were then lifted with trypsin-EDTA, centrifuged at 400 xg (5 min, 4°C), aspirated, then resuspended in 100 μ l of PBS. The concentrated cell solutions were transferred to EPR tubes. A 1 μ l aliquot was retained and diluted to obtain the number of cells in each solution.”

- The preclinical work has been performed in different mouse backgrounds: BALB/c and C57/Bl6 mice. The control experiments were performed in BALB/c mice. A justification for the use of the mice is required. How comparable are the models so that conclusions can be well supported? For instance, n =9 in this experiment (see biostatistical justification below).

The models for UO are commonly performed in C57/Bl6 mice (reference 2 in the supplemental information, but also <https://doi.org/10.1152%2Fajprenal.00384.2009>, and <https://doi.org/10.1038/labinvest.2014.50>) and our results around the model outcomes agreed with previous literature.

The FAN model has commonly been implemented in C57Bl/6 or CD1 mouse strains, however in our initial trials with this model, these mice suffered a very high degree of mortality (>80%) using doses indicated in the literature. We found an older literature source (reference 8 and 9 in the supplemental information) that showed BALB/c mice had were more robust to the FAN model with reduced mortality in the AKI phase. BALB/c has been shown to have more resistance to obstruction-mediated injuries and more reliably generates the CKD phenotype (reference 8 and 9 in the supplemental information). Switching to BALB/c mice, we found 100% survival after FAN implementation. We agree that the same strain would have been experimentally cleaner, however we were not comfortable with a model producing 80% mortality when switching strains allowed us to reduce animal numbers in the study.

In our work, the models are distinct, and our key endpoints are the relative changes in contrast enhancement, kinetics, and spatial distribution relative to baseline conditions, which was afforded by our longitudinal studies. We are encouraged that our imaging approach is not limited to a single model or strain of mice.

- Compared to standard renal excreted Gd-based CAs (Gadavist, Magnevist, Dotarem), how accurate is the GFR determination with glucoverdazyl? Also, authors should compare the performance of glucoverdazyl compared to verdazyl. What is the real benefit of using the linear gluconate compared to the other similar ORCAs (i.e., cytotoxicity is also meager with trityls and TEMPO derivatives, so what is special about this organic radical compared to the other stable ORCAs?)?

A 2021 book chapter dedicated to DCE-MRI for renal functional imaging had the following to say about what is known about GBCAs and GFR determination:

“Today, little is known about the potential of gadolinium-enhanced MRI for the assessment of the regional renal blood flow and the regional glomerular filtration rate. Nor do we know the limitations and accuracy of this technique under different pathological and pathophysiological conditions. Thirdly, we do not know the clinical applicability of DCE-MRI and the role it may have in clinical diagnosis. (https://doi.org/10.1007/978-1-0716-0978-1_12)”

The current manuscript initiates glucoverdazyl as a redox-stable ORCA towards defining the potential of DCE-MRI for imaging-based urography. By comparing our imaging results to a validated standard GFR technique (i.e. transdermal measurement), we have shown that our implementation of glucoverdazyl holds promise towards this goal.

While beyond the scope of the current work, we have already synthesized a variety of verdazyls with different pendant groups, ranging from galactose to cellobiose to C₆-linked carbohydrates, to amino acid-linked analogs. We are undertaking the evaluation of cell uptake and biodistribution of these contrast agents to answer the question posed by the reviewer: what is the effect of the pendant group on contrast agent performance? We are eager to answer this question.

The verdazyl radical is special relative to the other ORCAs reported because it takes advantage of radical delocalization for stability, rather than steric shielding like the trityl and TEMPO-derived radicals. The result is a chemically stable radical to the breadth of *in vivo* pH (see next question) and redox conditions that could otherwise negatively impact the other reported ORCAs.

- What is the effect of the pH on the EPR and MR signal (r1 relaxivity)?

In order to evaluate the effect of pH, we evaluated the EPR signal of glucoverdazyl at pH 3, 7, and 11 (Response Fig. 7A). There was no significant change in EPR signal between the pH points evaluated. The relaxation rate of the same solutions was measured (Response Fig. 7B). No change in relaxation rate was observed at pH 11 relative to pH 7, but a small ~33% increase at acidic pH 3 (Fig. 7B). This indicates that while the radical remains stable across a broad pH range, the rate of dissociation may change. This may lead to a change in perceived T₁.

Response Figure 7. Change in glucoverdazyl radical stability and relaxation by pH. A) Change in EPR activity of a 3 mM glucoverdazyl solution in PBS at pH 3, 7, or 11. B) Relaxation of a 3 mM glucoverdazyl solution in PBS at pH 3, 7, or 11. The change in relaxation compared to pH = 7 is shown as $\epsilon^* = \left(\frac{1}{T_{1HSA}} / \frac{1}{T_{1PBS}}\right)$.

We have included the following additions to the manuscript to indicate the results of these experiments, and have included the figures in the Supporting Information as indicated in the text below:

“In addition, glucoverdazyl was found to be stable at basic pH (i.e. pH 11) by both EPR and relaxation rate measurements, and demonstrating a small change in relaxation rate at pH 3 (ε*=1.33 relative to pH 7) (Supporting Fig. 13).”

We have added the following to the experimental section:

“The EPR was tuned to a sample of glucoverdazyl or TEMPO in PBS prior to any of the stability measurements. Once tuned, solutions of glucoverdazyl or TEMPO were prepared (20 mM in mouse serum, 5 mM in either 4 mM sodium ascorbate buffer (pH 7.4), 10 mM glutathione (GSH), 10 mM hydrogen peroxide (H₂O₂), or 3 mM in PBS of pH = 3, 7, or 11). A single spectrum was acquired and the peak height of the most intense peak for either compound was locked. EPR scans were then acquired every 5 s for 2 hr (mouse serum) or 1.5 hr (ascorbate, H₂O₂, glutathione, and pH) to measure percent change in the activity. For stability measurements of glucoverdazyl in water, a 5 mM sample was prepared and left in a fume hood exposed to light, or wrapped in tinfoil and left in a dark freezer at -20°C. Periodically, these solutions were sampled and measured by the EPR after it was tuned using a freshly prepared 5 mM sample of glucoverdazyl.”

“For relaxivity measurements, the same imaging phantom was used with contrast agent concentrations of 1 to 3 mM, 1 mM or 0.1 mM contrast agent in 0% to 22.5% HAS in PBS, or 3 mM contrast agent in PBS of pH 3, 7, or 11. Contrast agent concentration following imaging was verified by electron paramagnetic spectroscopy. To measure the longitudinal relaxation rate (R_1), an inversion recovery RARE sequence was implemented with the following parameters: Slice thickness of 5 mm, FOV of 50 × 50 mm, average = 1, matrix size = 96 × 96, $TE_{\text{eff}} = 17$ ms, TR = 5,000 ms, TI = 50, 75, 100, 150, 200, 250, 300, 400, 600, 800, 1200, 2400, and 4800 ms, and acquisition time of 2 minutes 30 seconds *per* TI. Longitudinal relaxation rates were extracted using the mapping MATLAB 2 routine written by J. Barral, M. Etezadi-Amoli, E. Gudmundson, and N. Stikov (2009), and modified by J. Rioux (2022). The longitudinal relaxivity (r_1) was extracted from the slope of the plot of $1/T_1$ vs. contrast agent concentration.”

- Figure 3, it is unclear how the authors calculated the RDTC maps. What can we conclude from the imaging?

We have rewritten the description of the method to describe RDTC maps as follows, and added this to the Supporting Information: Using the voxel-wise T_1 -weighted intensity values for each imaging time point during the ~45 min scan, a MatLab script was defined to extract the RDTC, k , as follows: A linear regression was fit to the natural logarithm (\ln) of the intensity values from $t = 2.5$ to 40 min, and the slope of this linear regression yielded the RDTC, k , *per* voxel (manuscript Fig. 2C). The k map was then overlaid on an anatomical reference image to yield the image-based maps of kidney function. An ROI was drawn over each kidney, or over the renal cortex and the medulla of each kidney. RDTC values shown in graphs are the mean of voxel-wise k values within an ROI.

Through our mapping of RDTC in healthy mice and given the observed consistency, we can conclude that any deviation in RDTC values from those observed in healthy mice is indicative of kidney dysfunction. Since RDTC is a measurement of how quickly glucoverdazyl moves through the kidney, the more positive this value becomes indicates a decrease in contrast clearance through the kidney. The ability to map RDTC by voxel in the kidney allows us to determine where and to what extent this dysfunction is occurring.

- The justification for the mismatch between the GFR determined by transdermal fluorescence and dynamic contrast-enhanced magnetic resonance imaging (DCE-MRI) in folic acid nephropathic (FAN) mice (Figure 4) is not convincing. One should see a clear difference on days 0, 15, and 30, not only on days 0 and 15. What happened on day 30? Transdermal fluorescence seems to perform better in identifying those differences, even though statistically, the SD is huge. Can the authors elaborate on this?

The recovery from initial folic acid-induced fibrotic injury in mice is variable and can differ depending on the strain of mouse studied (<https://doi.org/10.1172/jci.insight.164626>). Our observations are consistent with other partially susceptible strains such as 1229SvlmJ which display an initial tubule degeneration, interstitial inflammation, and fibrosis but no tubule dilation with recovery within 28 days.

Evaluating the histological outcomes of the FAN model in BALB/c mice, we see a clear increase in fibrosis on Day 15 in both the cortex and the medulla, but a reduction in fibrosis in both renal regions at Day 30 (manuscript Fig. 4C). In fact, the reduction in cortical fibrosis is greater than that observed in the medulla. With fibrosis as a marker of disease severity, Day 30 in our FAN model may represent an intermediary stage between AKI and CKD. The RDTc maps provided in the current manuscript recapitulate this finding: the largest change in RDTc at day 30 is in the medullary region. This reduction in disease severity was also recapitulated in a separate publication using multiparametric MRI in mice (10.1152/ajprenal.00128.2018). In this work, GFR was evaluated by DCE-MRI using gadodiamide as contrast agent 2 and 4 weeks after FAN induction, revealing a similar trend of return towards baseline of GFR, in agreement with the current work.

In another 2018 publication (10.1002/mrm.26955), Lilach Lerman describes an advantage to directly imaging kidney function at the level of the kidney as opposed to the reliance upon blood-based sampling procedures, which is the ability to define the contrast agent kinetics in the medulla (called the inner medullary papilla in this publication, or IMP). The kinetics in the IMP relate to the kidney outflow function, which Dr. Lerman states "...may allow for a more complete depiction of renal functional parameters.". This statement parallels our own justification for the difference between the glucoverdazyl values and the transdermal values of GFR as illustrated in manuscript Fig. 5D, where the impact of day 15 fibrosis appears to reduce cortical filtration, but the fibrosis persisting at day 30 impacts the outflow function, which we call "Excretion". While this reasoning is a proposal based on the data we currently have at hand, it will be important to elaborate on this investigation in the future in order to more fully reconcile the effect of decoupling kidney outflow function from filtration. In agreement with Dr. Lerman, it appears that GFR mapping in the kidney takes advantage of this decoupling, which is not afforded to transdermal measurements, and may provide a deeper granularity to renal disease diagnosis.

- The calculation of GFR and RDTC is not well described. It seems that the authors calculated the first decay with a pseudo-first order inverse kinetic model, but the clearance combines a second-order kinetic value originating from the accumulation of probes in the diseased models. How was this accounted for? A linear fitting will not serve this purpose, I am afraid. Could this explain the mismatch between transdermal fluorescence and dynamic contrast-enhanced magnetic resonance imaging?

This comment is well taken and relates directly to the publication by Dr. Lerman cited above (10.1002/mrm.26955). Here, the model, which is the classical mathematical view of kidney function, comprises two first-order terms: one describing filtration in the kidney (manuscript Fig. 5D), and one describing flow out of the kidney ('Excretion' in manuscript Fig. 5D). In arriving at our method of kinetic analysis, we also considered the analytics applied to the transdermal approach. When investigating this method as reported in 10.1152/AJPRENAL.00279.2012, we noted that GFR was calculated from the rate constant of the excretion phase of the clearance curve. The authors assert that their was no significant difference when using a 1- or 2-compartment model. The same outcomes was found by a second group (<https://www.ncbi.nlm.nih.gov/pmc/articles/PMC7341488/>). This single exponential readily linearizes following semi-log formatting, simplifying the two-compartment model.

- Authors should tone down some of the statements in the text. For instance:
o "Regardless of the Gd-based contrast agent used, ~20% of the injected dose is deposited irreversibly in the body, including the bone and brain." Rather, 1-2% of patients were injected continuously with labile Gd-complexes for several days. For stable Gd-complexes, Gd is sometimes found in the brain at ppm levels in the form of phosphonates (Radiology: Cardiothoracic Imaging 2019; 1(3):e190104 and J. Magn. Reson. Imaging 2009;30:1259–1267)

We appreciate the reviewer pointing this out and we have changed the text as follows in order to properly convey the limited risk of GBCAs:

"Regardless of the Gd-based contrast agent used, however with macrocyclic agents being more stable to linear chelators, there is a risk that low levels of gadolinium can be retained, for example at ppm-levels as phosphonates in the brain (Radiology: Cardiothoracic Imaging 2019; 1(3):e190104 and J. Magn. Reson. Imaging 2009;30:1259–1267), and be slowly cleared over months to years."

Conclusions:

- The data do not support some conclusions/findings. For example:"...allowing the acquisition of quantitative and qualitative kidney functional information." The imaging provided is rather qualitative, not quantitative. The calculated kinetic values are relative to the dose, perfusion, and proton-base catalyzed events. This is a strong declaration that is not supported at all by the data provided.

We agree the imaging provided in this manuscript is of qualitative value and that the kinetic data acquired is relative to dose, perfusion, and proton-base catalyzed events. We have modified the text by removing "quantitative and qualitative".

Scholarly presentation:

- The presentation of the data should be improved in most figures. The letters are too small in the labels and legends from the graphs and plots from the central figures (practically unreadable).
- The resolution of figures S12-S14 is low.

All figure text was resized and increased in quality to improve readability.

Appropriate use of statistics and treatment of uncertainties:

- The biostatistics methods and analysis are not well justified. Authors should comment on the sample sizes chosen to ensure adequate power for the predicted effect size. So, how exactly the authors determined $n=9$ for the in vivo experiments?

Using the given error rates and effect sizes, and considering an $\alpha=0.05$ and $1-\beta=0.8$, we calculated that the minimum number of samples *per* group to be 4 for FAN and UO analyses. The $n=5$ utilized in this work satisfies these power needs. The use of a two-way ANOVA allowed multiple group comparisons over a repeated measures design, evaluating for both treatment and time effects without compounding measurement error.

When we determined our conversion factor to transform RDTc to GFR, we wanted to use a large value as n so as to evaluate the variability inherently associated with the baseline measurement. However, given the consistency in healthy mouse RDTc, we chose a smaller value of n for diseased models in order to reduce animal number.

We have added a Statistics section (2.17) to the supplemental information.

- The authors should provide more information about the following:
 - o the expression of the size effect.
 - o comparison of the groups, and matched conditions. For instance, fasted animals, *ad libitum*, glucose levels, weight, etc.
 - o Assumption for the normality and paired data.
 - o the appropriateness of the statistical test selected for this work.

We have elaborated on the animal model section, adding the following to section 2.15 of the SI: All studies were initiated when animals were 9-10 weeks of age. All animals were housed in conventional breeding cages, fed standard rodent chow and received water *ad libitum*, and were housed in a facility with a 12-hour light cycle.

The use of a two-way ANOVA allowed multiple group comparisons over a repeated measures design, evaluating for both treatment and time effects without compounding measurement error. All analyses were performed in GraphPad Prism, and included tests for assumption of normality and equality of variances. In all cases, the assumptions were upheld by the data, otherwise non-parametric analyses would have been performed. The statistical methods have been updated in the SI with the following statement: Prior to group-wise comparisons, all data was evaluated for assumptions of normality and equality of variances.

Reviewer #2 (Remarks to the Author):

Summary

The authors demonstrate use of glucoverdazyl for the assessment of GFR by MRI in mice with acute kidney injury (surgical or medical) and chronic kidney disease (at 15 and 30 days).

Strengths:

1. This study tests a potentially novel verdazyl for use at MRI as a novel contrast agent for assessing GFR. In current state, the methods used for assessing GFR have some limitations.

Key Weaknesses:

1. The rationale and background for the method deserve additional support and argumentation in comparison to current state with respect to clinical applicability and usability of spatial information, and to the pragmatic application of MRI-based methods for assessing GFR.

We agree that the previous version of the manuscript was able to be improved regarding establishing the clinical need for imaging-based GFR assessment. We have amended the text to better discuss this (see page 2 of the manuscript) as follows:

“Imaging-based GFR assessment is of particular interest because it allows for a direct link between structural alterations (i.e. renal artery stenosis, ureteral obstruction) and changes in kidney filtration. In addition to this, GFR may be measured at the single kidney level. This is of particular interest when monitoring patients after partial nephrectomy or after living kidney donation where early increases in single kidney GFR are predictive of beneficial outcomes ([10.1016/j.kint.2022.01.034](https://doi.org/10.1016/j.kint.2022.01.034)). “

2. In current state, DCE-MRI using existing molecules is feasible and safe, even in patients with kidney disease.

The reviewer is correct in stating that GBCAs are, overall, a safe contrast agent used for decades. The major risk of NSF related to use of GBCA in acute and chronic kidney disease is related to use of older (American College of Radiology [ACR]) Group 1 agents and is very low with ACR Group 2 agents estimated to be near zero (<https://doi.org/10.1001/jamainternmed.2019.5284>). Nevertheless, the risk of NSF remains near zero and not zero and there is a possibly historical but pervasive bias in the medical community to continue to avoid GBCA in patients with altered renal function (once bitten twice shy) which may take years to undo. Similarly, unknown consequences of gadolinium retention after GBCA use, which may be raised in patients with renal disorders or dysfunction, has given the medical imaging community pause and search for alternative lower dose or higher stability GBCA (<https://doi.org/10.1097/RLI.0000000000000944>). Given these points, we feel that the development of a non-gadolinium-based contrast agent for DCE-MRI will be of clinical interest.

Reviewer 1 also had some concerns regarding GBCA safety and the text has been modified as indicated in our responses # to Reviewer 1. We hope these will also satisfy the concerns of Reviewer 2.

“Regardless of the Gd-based contrast agent used, however with macrocyclic agents being more stable to linear chelators, there is a risk that low levels of gadolinium can be retained, for example at ppm-levels as phosphonates in the brain (Radiology: Cardiothoracic Imaging 2019; 1(3):e190104 and J. Magn. Reson. Imaging 2009;30:1259–1267), and be slowly cleared over months to years.”

Specific comments

Title

1. OK

Abstract

1. Sentence 2: Revisions to eGFR formulae (implemented in the last 2-3 years) and now used in clinical practice have omitted race from their designations.

We agree with the reviewer that more recent eGFR formulae are derived from more representative populations. Indeed, a variety of eGFR formulae may be employed in practice. We have revised the sentence to acknowledge that only certain formulae are derived from under-represented populations.

“Glomerular filtration rate (GFR) is the gold standard measure of kidney function, which is estimated by entering hematological, physiological and demographic information into a variety of available equations, where some have been derived from an under-representative population.”

2. Sentence 3: The clinical value of spatial information within a kidney as it relates to kidney function should be alluded or specified.

We agree with the reviewer that the value of spatial information should be specified here and have revised the text accordingly.

“This estimated GFR provides no spatial information about kidney dysfunction, which could be of particular interest when monitoring patients after partial nephrectomy or after living kidney donation.”

3. In current practice, kidney function assessment is by scintigraphy. The rationale for replacing it with DCE-MRI should be stated.

We agree with the reviewer that justification for the use of DCE-MRI over scintigraphy. We have modified the abstract as follows:

“Dynamic contrast enhanced magnetic resonance imaging (DCE-MRI) could be added to the anatomical imaging already performed, resulting in a powerful ‘one stop shop’ for renal assessment in cases of suspected AKI and CKD,…”

However, given the strict word limitations for the abstract we have specifically addressed this comparison to scintigraphy on page 2-3 of the main manuscript.

“Camera-based imaging of GFR is, however, currently limited to ^{99m}Tc -diethylenetriaminepentaacetic acid (DTPA) single photon emission computed tomography (SPECT). ^{99m}Tc -DTPA imaging suffers from two sources of error leading to a wider confidence interval of the determined GFR relative to eGFR: background subtraction necessary for the correct measurement of percentage injected dose, and the estimation of renal depth from a population-derived nomogram based on patient height and weight in order to correct for signal attenuation.^[14,15] Additionally, SPECT is also associated with limited spatial resolution and structural detail, however which is achievable by other imaging modalities such as magnetic resonance imaging (MRI).^{[16,17]”}

Introduction

1. Line 46: Please specify the interventions that are possible once CKD is diagnosed to prevent progression.

We have added two current interventions for CKD to the text.

“CKD outcomes are improved with early interventions, such as renin-angiotensin system blockade (10.1056/NEJM19931113292004, 10.1016/S0140-6736(98)10363-X, 10.1056/NEJMoa011161) or canagliflozin-mediated sodium-glucose cotransporter inhibition (10.1056/NEJMoa1811744), necessitating earlier detection^[5].”

2. Line 55: It is recognized that eGFR is biased relative to true GFR and is especially biased in formulas that were derived from biased populations. More accurate tests that incorporate cystatin C are available.

We agree with this statement and have modified the text accordingly.

“These issues may be partially mitigated by incorporation of cystatin C into the equation, however eGFR equations also assume steady-state creatinine and cystatin c levels and do not account for alterations in or alternate routes of creatinine production, exacerbating variability.”

3. Line 58: It is an overstatement to say it is “imperative”, but it certainly would bring value to have more accurate and easily-obtained methods. MRI-based methods of measuring GFR probably are not scalable to the general population or usable for screening; therefore, they do not probably qualify as easily obtained. In other words, even if it is perfectly accurate, from a practical standpoint, obtaining MRI in all patients who otherwise would have eGFR measured is not feasible. Essentially all patients who come to the ED or are inpatients undergo eGFR testing. In addition, many outpatients with risk factors also undergo testing. It is extremely common. The commonality is problematic because any test that isn't easy, cheap, and broadly accessible isn't going to work as a substitute in the vast majority of patients. As a partial analogue to this, cystatin C is much more accurate than serum creatinine, but lack of availability and familiarity has limited uptake. Translate that same idea to MRI and it is orders of magnitude more challenging / infeasible.

We agree with the reviewer that the use of MRI-based methods of measuring GFR are likely not scalable to the general population. Thus, our approach, and indeed all imaging-based approaches, are likely to be employed in specific cases. As a result, we have changed ‘imperative’ to ‘important’. A good example of this is the monitoring of single kidney GFR post kidney donation. A recent manuscript by van der Weijden *et al*

suggested that assessment of single kidney GFR can predict long term kidney function in this population (<https://doi.org/10.1016/j.kint.2022.01.034>).

Another specific application is in potential renal donors or patients being considered for nephrectomy for other reasons. CT or MRI are typically performed in these patients to evaluate for anatomy (e.g. number, length, pathology of renal arteries, renal vein and collecting system variants) and pathologies (e.g. renal calculi or masses). In a separate session, patients undergo scintigraphic assessment of the kidneys to determine split renal function. MRI could serve as a 'one stop shop' to evaluate both anatomy, pathology and renal function thus increasing convenience for the patient and reducing cost and delays.

4. Line 61: It is stated that spatial information is "important information" for patients with CKD. This theme is repeated in the manuscript without explanation for how it would be helpful. There are not current clinical paradigms that make significant use of or would require this. Please explain how within-kidney spatial information is helpful or will be helpful in the future.

In response to this point, we have considered our response to comment 3 and the first key weakness described from this reviewer. We have modified the text as indicated in the response to the first key weakness raised, repasted below:

"Imaging-based GFR assessment is of particular interest because it allows for a direct link between structural alterations (i.e. renal artery stenosis, ureteral obstruction) and changes in kidney filtration. In addition to this, GFR may be measured at the single kidney level. This is of particular interest when monitoring patients after partial nephrectomy or after living kidney donation where early increases in single kidney GFR are predictive of beneficial outcomes (10.1016/j.kint.2022.01.034)."

5. Line 68: When describing the potential limitations of scintigraphy, please include both DTPA and MAG-3 (MAG-3 being more accurate than DTPA in patients with CKD or AKI). In the current description, only DTPA is included. Also, rather than a just a listing of potential weaknesses of scintigraphy, please provide specific numeric accuracy and precision data for both scintigraphic methods (with reference standards) in patients with kidney disease so the reader knows what the current state is that MRI is competing against.

We have included a description of MAG3, as well as specificity and accuracy data for imaging-based diagnosis of renal failure by ^{99m}Tc-DTPA as follows:

"The specificity and sensitivity of camera-based GFR as diagnostic for renal failure was 100% and 47.5% for ^{99m}Tc-DTPA (<https://doi.org/10.5455%2Faim.2017.25.99-102>). ^{99m}Tc-mercaptoazyltriglycerine (MAG3) is another SPECT-based method for imaging-based urography that is primarily cleared by tubular secretion, and is often used to evaluate renal plasma flow (14). However, clearance of this radiotracer corrected for body surface area correlates well with creatinine-based GFR measurements (10.2214/AJR.05.1025)."

6. Line 85: The greatest challenge for DCE-MRI in evaluation of GFR is not perceived risk of gadolinium (although I agree that is a barrier). Even if that issue is off the table,

the greatest challenge is pragmatic application of an expensive and time-consuming test as a replacement for pragmatic alternatives.

We agree that the perceived risk of gadolinium is only one barrier to clinical implementation. We would stress that we do not see our approach as a replacement for eGFR assessment which has proven utility for population-level screening. Nevertheless, there remain several clinical paradigms where eGFR is known to be inaccurate and alternative approaches (i.e. inulin clearance) are merited. In such cases we feel that our approach represents an appealing option. The key advantage is a one stop shop approach where both functional and anatomical data can be obtained in a single MRI session. For example, when renal function is down in at transplant, the renal artery (for stenosis) or vein (for thrombosis) can be evaluated at the same time as GFR with the method presented herein. The value of a one stop shop approach has been looked at extensively with CT already (<https://www.ncbi.nlm.nih.gov/pmc/articles/PMC6753513/>) and we feel glucoverdazyl could open the possibility to the same value for MRI.

7. Line 90: Please include in your discussion of NSF the differentiation of various types of GBCM. Some agents (gadobenate, macrocyclics) have extremely low risk of NSF, whereas other agents are high risk.

We agree with this suggest and have modified our text as follows:

“There is an increased risk of NSF in patients with severe renal dysfunction (AKI, dialysis patients, and stage 5 CKD) where lower rates of Gd clearance lead to higher residence times of the contrast agents.^[26] This risk appears to be mitigated in group II GBCA, as defined by the American College of Radiology (<https://doi.org/10.1001/jamainternmed.2019.5284>).”

8. Line 96: It is not accurate to say that 20% of the dose is deposited “Irreversibly” in tissues. The fraction of retained gadolinium is much smaller (<6%) than that, the fraction that is retained is slowly cleared over months or years, and the likelihood of retention is highly dependent on the agent (macrocyclic agents are retained a further order of magnitude less than linear agents).

We appreciate the reviewer pointing this out and we have changed the text as follows in order to properly convey the limited risk of GBCAs:

“Regardless of the Gd-based contrast agent used, however with macrocyclic agents being more stable to linear chelators, there is a risk that low levels of gadolinium can be retained, for example at ppm-levels as phosphonates in the brain (Radiology: Cardiothoracic Imaging 2019; 1(3):e190104 and J. Magn. Reson. Imaging 2009;30:1259–1267), and be slowly cleared over months to years.”

9. Line 131: (re: “toxicological concerns”): Modern GBCAs are incredibly safe and extensively tested. For modern agents, there is a tiny theoretical risk of NSF (0 to 0.07%), a rare risk of severe contrast reaction (less than 1 in 100,000), and common but

tiny amounts of gadolinium retention that are of unclear and doubtful significance in most patients. New agents will have a very high bar to pass to replace them because of the extensive testing that would be involved. During that extensive testing, it is extremely likely that rare side effects will emerge just as they did for gadolinium-based media. In the present state, contrast-enhanced MR based kidney perfusion and quantification can be performed with modern GBCAs. The reason it isn't done is not truly because of risk of NSF or retention. It is because there are more practical ways of acquiring the information, and there is not a present dominant clinical need for spatial kidney function data.

We appreciate the careful consideration of GBCA safety pointed out by the reviewer. Following along from the previous comment, we have removed "alleviate toxicological concerns" from the identified statement, editing it to the following in order to tone down safety concerns:

"The present work lays the ground for a novel class of ORCAs with the potential to provide for contrast enhanced-MRI in patients with renal dysfunction, and for the mapping of GFR onto the kidneys independent of the demographic characteristics of the patient."

Results and Discussion

2. Line 205-210 & Figure 2: Enhancement is observed in muscle and liver, even up to 40 mins post injection.

This increase in contrast enhancement in these tissues is a result of vascular perfusion of these tissues. The enhancement observed in these tissues is not significantly different relative to time 0, and do not significantly change in intensity immediately post injection relative to the end of the scan.

3. Line 343: Why did the fibrosis decrease on day 30? This seems very unusual to have rapid decline in fibrosis.

The recovery from initial folic acid-induced fibrotic injury in mice is variable and can differ depending on the strain of mouse studied (<https://doi.org/10.1172/jci.insight.164626>). Our observations are consistent with other partially susceptible strains such as 1229SvlmJ which display an initial tubule degeneration, interstitial inflammation, and fibrosis but no tubule dilation with recovery within 28 days <https://doi.org/10.1172/jci.insight.164626>.

4. Line 374: What is the reference standard on which transdermal fluorescence monitoring was validated? Further, has it been validated at day 0, day 15, and day 30?

The transdermal fluorescence measure is a commercial device developed and validated by MediBeacon Inc., and has been developed over a series of publications listed below. The device and FITC-sinistrin have been benchmarked against traditional blood sampling-based GFR measurements. The performance of the technology has been advanced to clinical trials for human use (<https://www.medibeacon.com/science/clinical/>).

- J. Friedemann, R. Heinrich, Y. Shulhevich, M. Raedle, L. William-Olsson, J. Pill, D. Schock-Kusch, *Kidney Int* **2016**, *90*, 1377–1385.
- L. Scarfe, D. Schock-Kusch, L. Ressel, J. Friedemann, Y. Shulhevich, P. Murray, B. Wilm, M. de Caestecker, *J Vis Exp* **2018**, *2018*, 58520.
- A. Shmarlouski, D. Schock-Kusch, Y. Shulhevich, V. Buschmann, T. Rohlicke, D. Herdt, M. Radle, J. Hesser, D. Stsepankou, *IEEE Trans Biomed Eng* **2016**, *63*, 1742–1750.
- D. Schock-Kusch, S. Geraci, E. Ermeling, Y. Shulhevich, C. Sticht, J. Hesser, D. Stsepankou, S. Neudecker, J. Pill, R. Schmitt, A. Melk, *PLoS One* **2013**, *8*, e71519.
- A. Schreiber, Y. Shulhevich, S. Geraci, J. Hesser, D. Stsepankou, S. Neudecker, S. Koenig, R. Heinrich, F. Hoecklin, J. Pill, J. Friedemann, F. Schweda, N. Gretz, D. Schock-Kusch, *Am J Physiol Renal Physiol* **2012**, *303*, DOI 10.1152/AJPRENAL.00279.2012.

5. Line 398: Please do not speculate about trends that are not statistically significant.
We have removed the indication of non-significant results.

Conclusion

1. There is substantial reference in the manuscript about the use of the technique for spatial mapping of kidney dysfunction. However, the methods and results presented to not assess kidney function differently within spatial regions of individual kidneys.

Regional changes to kidney function are indicated in Figs. 3 and 4 in the manuscript, and regional analysis (in this case cortex versus medulla and renal pelvis) are provided in the supporting information). Key regional difference in RDTC were identified between the cortical and medullary regions of the kidneys, and these differences were also recapitulated in the histological evaluation of FAN over time. Furthermore, our spatial analysis allowed for discrete evaluation of ligated and contralateral kidney function in the UUO model of disease.

References

1. Please include modern references 2021-2023 that discuss use of alternative methods of kidney function assessment by blood tests that do not use race and are now widely used in clinical care.

We have updated the references used throughout the manuscript.

Reviewer #3 (Remarks to the Author):

The authors describe the use of a novel organic radical based MRI contrast agent as a way to examine kidney function using a mouse model. The work is significant because of the established toxicity of standard gadolinium contrast agents in individuals with impaired kidney function.

Important results that form part of this paper include the establishment of the relative non-toxicity of this class of contrast agents, their relatively long lifetime in vivo, and their efficacy as contrast agents.

The methodology regarding the synthesis and characterization of the contrast agent is sound; the provided data establishes the identity of the material and there is certainly sufficient information to reproduce the synthesis. Establishing purity of stable free radicals is a little more challenging than many other organic compounds because they do not give sharp NMR spectra. The authors provide HPLC data that supports the purity of the samples, but additional information such as elemental analysis or UV-spectra (which can be compared with published data for the previously reported synthesis) would provide an even more solid foundation.

The radical and reduced forms of glucoverdazyl are well differentiated by HPLC and by UV-Vis spectroscopy, as show in supporting fig. 8 reproduced below. In its reduced form (compound 3), the solution is clear and colourless, absorbing only in the UV region. However, the installation of the radical induces a yellow colour, resulting in absorbance at 452 nm. From the HPLC trace at 210 nm, it can be seen that there is no detectable compound 3 present after compound 4 purification.

Supporting Figure 8. High performance liquid chromatography traces of compounds 3 and 4 to verify radical activity of the compound after radicalization step.

The reported cytotoxicity and animal studies appear to support the conclusions regarding toxicity and efficacy as a contrast agent; however, as a chemist less familiar with such biomedical studies I cannot comment on this aspect of the paper in more detail.

In general though, the significance of this study, in terms of breaking new ground for MRI contrast agents, makes it well worth publication in Nature Communications

REVIEWERS' COMMENTS

Reviewer #1 (Remarks to the Author):

The authors addressed all the remarks and the work should be considered for publication.

Minor comments:

- The authors should carefully revise the Supplementary Information (SI) to correct any typos and ensure consistency in the nomenclature used (e.g. HSA instead of HAS).
- I suggested that the authors consider incorporating a sentence discussing the limitations of their work. Including such information would enhance the overall discussion and provide valuable scientific perspectives.

Reviewer #2 (Remarks to the Author):

Thank you for making the requested changes to the manuscript. I have no further questions or recommendations.

Reviewer #3 (Remarks to the Author):

The authors have provided a modified manuscript that adequately addresses my concerns and provides significantly more detail as to the stability of glucoverdazyl towards oxidants and reductants commonly present in vivo. My area of expertise does not cover the biomedical aspects of the paper and so I cannot comment further on those aspects; however if the authors response to the biomedical aspects is satisfactory then I consider the paper suitable for publication.